# Genetic and chemical validation of *Plasmodium falciparum* aminopeptidase *Pf*A-M17 as a drug target in the hemoglobin digestion pathway

Rebecca CS Edgar[1,2], Ghizal Siddiqui[3], Katheryn Hjerrild[1], Tess R Malcolm[4,5], Natalie B Vinh[6], Chaille T Webb[4,5], Clare Holmes[7], Christopher A MacRaild[3], Hope C Chernih[1,2], Willy W Suen[7], Natalie A Counihan[1,2], Darren J Creek[3], Peter J Scammells[6], Sheena McGowan[4,5], Tania F de Koning-Ward[1,2]*

[1]School of Medicine, Deakin University, Geelong, Australia; [2]The Institute for Mental and Physical Health and Clinical Translation, Deakin University, Geelong, Australia; [3]Drug Delivery, Disposition and Dynamics, Monash Institute of Pharmaceutical Sciences, Monash University, Parkville, Australia; [4]Biomedicine Discovery Institute and Department of Microbiology, Monash University, Clayton, Australia; [5]Centre to Impact AMR, Monash University, Melbourne, Australia; [6]Medicinal Chemistry, Monash Institute of Pharmaceutical Sciences, Monash University, Parkville, Australia; [7]CSIRO Australian Centre for Disease Preparedness, Geelong, Australia

**Abstract** *Plasmodium falciparum,* the causative agent of malaria, remains a global health threat as parasites continue to develop resistance to antimalarial drugs used throughout the world. Accordingly, drugs with novel modes of action are desperately required to combat malaria. *P. falciparum* parasites infect human red blood cells where they digest the host's main protein constituent, hemoglobin. Leucine aminopeptidase *Pf*A-M17 is one of several aminopeptidases that have been implicated in the last step of this digestive pathway. Here, we use both reverse genetics and a compound specifically designed to inhibit the activity of *Pf*A-M17 to show that *Pf*A-M17 is essential for *P. falciparum* survival as it provides parasites with free amino acids for growth, many of which are highly likely to originate from hemoglobin. We further show that loss of *Pf*A-M17 results in parasites exhibiting multiple digestive vacuoles at the trophozoite stage. In contrast to other hemoglobin-degrading proteases that have overlapping redundant functions, we validate *Pf*A-M17 as a potential novel drug target.

**\*For correspondence:**
taniad@deakin.edu.au

**Competing interest:** The authors declare that no competing interests exist.

## Editor's evaluation

This article offers further characterization of PfA-M17, a *P. falciparum* aminopeptidase that has been studied for some years and was previously shown to be an essential protein predicted to function in late steps of hemoglobin hydrolysis by erythrocytic parasites. The new report adds valuable demonstration of impacts of PfA-M17 knockdown, description of the synthesis and characterization of a novel PfA-M17 inhibitor with high nanomolar activity against cultured parasites, and metabolomic analysis of inhibited parasites adding evidence for hemoglobin to be the primary target of the protease. The work is important to our understanding of the roles of aminopeptidases in parasite biology and the claims are convincingly supported by the data.

**eLife digest** Malaria is a disease spread by mosquitoes. When infected insects bite the skin, they inject parasites called *Plasmodium* into the host. The symptoms of the disease then develop when *Plasmodium* infect host red blood cells. These parasites cannot make the raw materials to build their own proteins, so instead, they digest haemoglobin – the protein used by red blood cells to carry oxygen – and use its building blocks to produce proteins.

Blocking the digestion of haemoglobin can stop malaria infections in their tracks, but it is unclear how exactly *Plasmodium* parasites break down the protein. Researchers think that a group of four enzymes called aminopeptidases are responsible for the final stage in this digestion, releasing the amino acids that make up haemoglobin. However, the individual roles of each of these aminopeptidases are not yet known.

To start filling this gap, Edgar et al. set out to study one of these aminopeptidases, called PfA-M17. First, they genetically modified *Plasmodium falciparum* parasites so that the levels of this aminopeptidase were reduced during infection. Without the enzyme, the parasites were unable to grow. The next step was to confirm that this was because PfA-M17 breaks down haemoglobin, and not for another reason. To test this, Edgar et al. designed a new molecule that could stop PfA-M17 from releasing amino acids. This molecule, which they called 'compound 3', had the same effect as reducing the levels of PfA-M17. Further analysis showed that the amino acids that PfA- M17 releases match the amino acids found in haemoglobin.

Malaria causes hundreds of thousands of deaths per year. Although there are treatments available, the *Plasmodium* parasites are starting to develop resistance. Confirming the role of PfA-M17 provides a starting point for new studies by parasitologists, biologists, and drug developers. This could lead to the development of chemicals that block this enzyme, forming the basis for new treatments.

## Introduction

Malaria is an infectious disease caused by protozoan parasites belonging to the *Plasmodium* genus, of which *Plasmodium falciparum* is the deadliest to humans. In 2020, there were more than 600,000 deaths attributed to malaria infections, the majority of these occurring throughout sub-Saharan Africa and South-East Asia (**WHO, 2021**). The current front-line therapeutic artemisinin and its derivatives, as well as partner drugs used in combination therapies, are under threat as resistance to these drugs continues to arise. Artemisinin resistance, which was once localized to the Greater Mekong sub-region within Asia, has now been identified elsewhere, including most worryingly in malaria endemic regions of Africa, with *de novo* resistance identified in Rwanda and delayed parasite clearance times confirmed in Uganda (**Uwimana et al., 2020**; **Uwimana et al., 2021**). The threat of a growing spread of resistance highlights the need to identify new therapeutic targets and compounds with novel modes of action (**Ataide et al., 2017**).

The intra-erythrocytic cycle of *P. falciparum* is responsible for the clinical manifestations of disease and is the target of most antimalarial drugs. Parasite survival during the erythrocytic stage is dependent upon the digestion of host hemoglobin to provide amino acids essential for parasite growth, with the exception of isoleucine which is absent from human hemoglobin and, therefore, has to be taken up from the extracellular environment (**Babbitt et al., 2012**). Digestion of hemoglobin also creates space in the erythrocyte to accommodate the growing parasite, as well as providing a mechanism to regulate the osmotic pressure of the host cell (**Krugliak et al., 2002**; **Lew et al., 2004**; **Liu et al., 2006**; **Rosenthal, 2002**). Hemoglobin digestion begins during the early ring stage of growth, but formation of hemozoin crystals, a detoxified version of the digestive by-product heme, can only be visualized by microscopy in the more developed trophozoite stage in a specialized acidic compartment termed the digestive vacuole (DV) (**Abu Bakar et al., 2010**). The DV contains an array of proteases responsible for the release of peptides from hemoglobin, including plasmepsins, falcipains, falcilysin, and aminopeptidases **Goldberg, 2005**; it is the latter which are speculated to perform the final step of amino acid cleavage from the amino-terminal end of these peptides (**Gavigan et al., 2001**).

As aminopeptidases are implicated in the final step of hemoglobin digestion, they are promising novel therapeutic targets. The *P. falciparum* genome encodes two neutral metallo-aminopeptidases: the M1 alanyl aminopeptidase (*Pf*A-M1; *Pf*M1AAP) and M17 leucyl aminopeptidase (*Pf*A-M17;

*Pf*M17LAP). The broad-spectrum aminopeptidase inhibitor bestatin has been shown to kill *P. falciparum*, with parasites overexpressing *Pf*A-M17 displaying resistance to this drug, suggesting *Pf*A-M17 is its target *in vivo* (*Gardiner et al., 2006*; *Nankya-Kitaka et al., 1998*). *Harbut et al., 2011* also showed that treatment of *P. falciparum* with bestatin reduced hemoglobin digestion and decreased isoleucine uptake. However, specific inhibition of *Pf*A-M17 using an activity-based probe based on the bestatin scaffold resulted in ring-stage arrest and parasite death, whilst an equivalent probe designed to specifically inhibit *Pf*A-M1 resulted in DV swelling and stalling of parasite growth much later at the trophozoite stage. This led the authors to conclude that *Pf*A-M17 may be playing a role outside of, or in addition to, hemoglobin digestion (*Harbut et al., 2011*). Several series of inhibitors designed to inhibit both *Pf*A-M17 and *Pf*A-M1 have also been developed and these suppress a range of *Plasmodium* species *in vivo* and *in vitro* (*Mistry et al., 2014*; *Skinner-Adams et al., 2012*; *Vinh et al., 2019*). *Drinkwater et al., 2016*, for example, developed a series of dual inhibitors that killed sensitive and multi-drug resistant parasites in the micromolar range, validating *Pf*A-M1 and *Pf*A-M17 as potential novel therapeutic targets.

*Pf*A-M17 is a 68 kDa cytoplasmic enzyme that forms a homo-hexamer in its active form, with optimal function at neutral pH, similar to the pH of the parasite cytoplasm (*Dalal and Klemba, 2007*; *Mathew et al., 2021*; *McGowan et al., 2010*). *Mathew et al., 2021* recently confirmed the cytoplasmic localization of *Pf*A-M17, which suggests that hemoglobin-derived peptides are exported from the DV into the parasite cytoplasm, either through the chloroquine-resistance transporter or by other unidentified mechanisms (*Shafik et al., 2020*). Once in the cytoplasm, it is believed that hemoglobin-derived peptides are digested by *Pf*A-M17; however, it is also possible that *Pf*A-M17 plays an additional role in the catabolic turnover of peptides from other origins. Both *Pf*A-M17 and *Pf*A-M1 are able to cleave single amino acids from the N-terminus of small peptide chains, with *Pf*A-M1 showing a broad substrate specificity with a preference for Met >Ala > Leu over the charged and/or polar residues Lys >Arg >> Gln (*Poreba et al., 2012*). In contrast, *Pf*A-M17 shows almost exclusive selectivity for leucine and tryptophan *in vitro*, with leucine being one of the most abundant amino acids in adult hemoglobin (*Hill et al., 1962*; *Poreba et al., 2012*). Functionally, leucine has been shown to be an important substrate of the isoleucine transporter at the parasite membrane (*Martin and Kirk, 2007*). As isoleucine is the only amino acid that is absent from hemoglobin and is sourced from the host serum, leucine generated by the parasite may be important for the uptake of isoleucine across the parasite membrane (*Babbitt et al., 2012*; *Liu et al., 2006*). While repeated attempts to knockout *Pfa-m17* have failed (*Dalal and Klemba, 2007*; *Zhang et al., 2018*), suggesting it is essential for intra-erythrocytic growth, its ortholog could be disrupted in both *Plasmodium berghei*, a rodent malaria species, and in the closely related apicomplexan parasite, *Toxoplasma gondii*, leading in both cases to a significant effect on parasite replication (*Lin et al., 2015*; *Zheng et al., 2015*).

In order to functionally characterize *Pf*A-M17 and determine its contribution to *P. falciparum* survival, we created a conditional knockdown parasite line that enabled *Pf*A-M17 expression to be regulated via a riboswitch (*Prommana et al., 2013*). This revealed that parasites depleted of *Pf*A-M17 experience a growth delay and fail to expand in culture, subsequently leading to parasite death. We have also designed, synthesized, and characterized a novel and specific small molecule inhibitor to *Pf*A-M17, compound **3**, and parasites treated with this inhibitor demonstrate a similar growth phenotype to *Pf*A-M17 knockdown parasites. We also found that loss of *Pf*A-M17, either by knockdown or treatment with **3**, resulted in multiple hemozoin fragments that were representative of multiple digestive vacuoles within trophozoite stage parasites. Finally, metabolomic analysis of knockdown and **3**-treated parasites revealed that peptides accumulating after the depletion of *Pf*A-M17 are likely to originate from hemoglobin, indicating that *Pf*A-M17 plays a role in the final stages of its digestion.

## Results

### Epitope tagging and incorporation of a *glmS* ribozyme into the *Pfa-m17* locus

To tease out the function of *Pf*A-M17 and determine its essentiality, we sought to use reverse genetics to deplete its expression using a conditional riboswitch system. Accordingly, the *Pf*A-*m17* locus was targeted by transfecting *P. falciparum* 3D7 with a pM17-HAglmS construct (*Figure 1A*). Transfectants underwent three rounds of drug cycling with WR99210 before a pure population of integrated

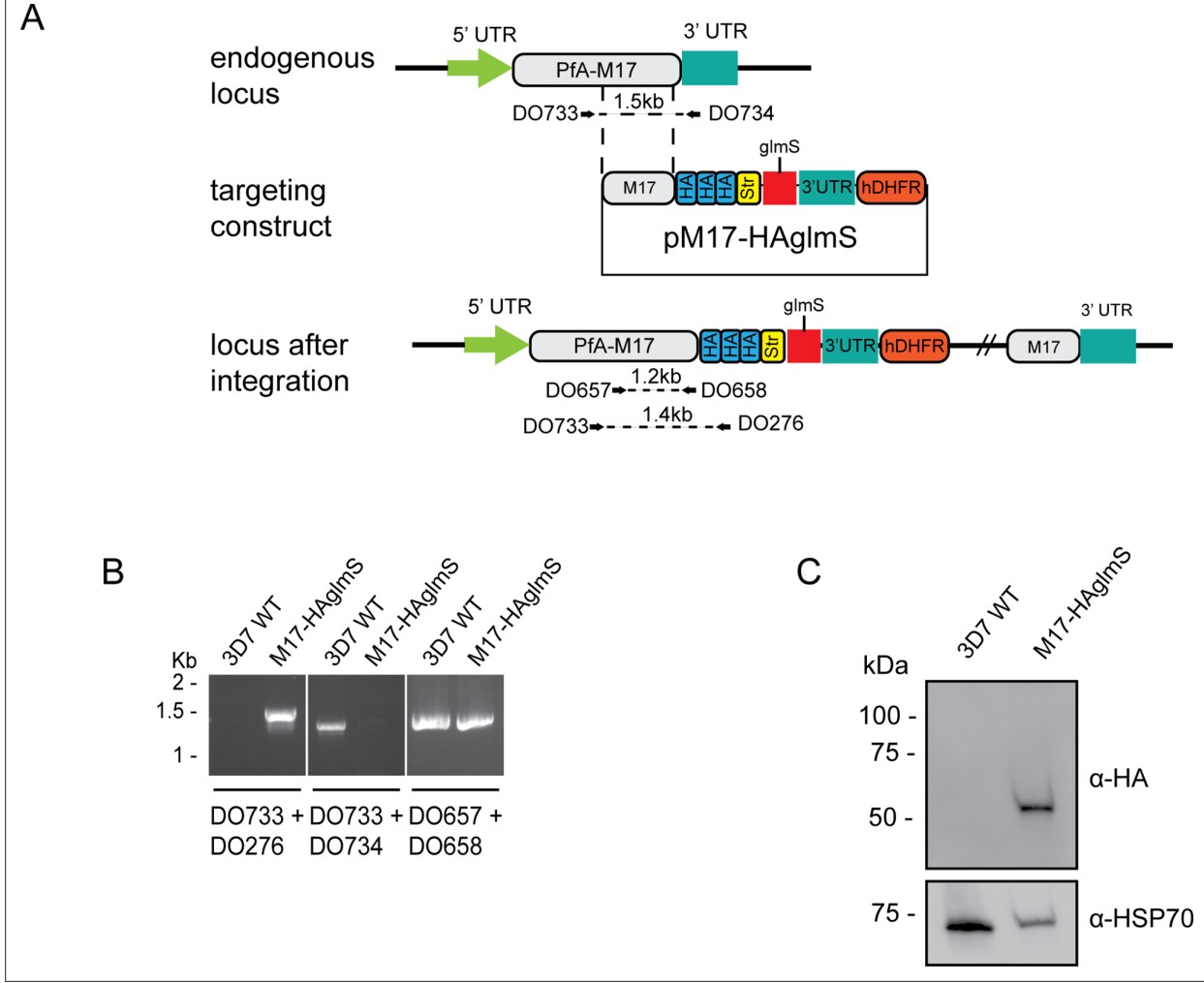

**Figure 1.** Generation of HA-tagged *Pf*A-M17-HAglmS transgenic parasites. (**A**) Schematic of the *Pfa-m17* locus, and locus after single crossover recombination with pM17-HAglmS. The pM17-HAglmS plasmid contained the last kilobase of the coding sequence excluding the stop codon fused in frame to 3 x haemagglutinin (HA) and a single strep II (Str) tag. The plasmid also includes a *glmS* ribozyme with a synthetic untranslated region (UTR) and the selectable marker human dihydrofolate reductase (hDHFR). Arrows indicate oligonucleotides used in diagnostic PCRs as well as their expected sizes. (**B**) Diagnostic PCR showing integration of pM17-HAglmS at the endogenous locus. PCR was performed using the oligonucleotide pairs outlined in (**A**) on DNA extracted from parasites before (*Pf*3D7) or after (*Pf*A-M17-HAglmS) transfection with the targeting construct. Oligonucleotides DO657 and DO658, which recognize the endogenous locus, serves as a positive control. (**C**) Western blot analysis of parasite lysates confirming HA expression. The predicted molecular mass of *Pf*A-M17-HA is 72 kDa, and HSP70 serves as a loading control.

The online version of this article includes the following source data and figure supplement(s) for figure 1:

**Source data 1.** Original gel electrophoresis presented in *Figure 1B* (panels 1 and 2).

**Source data 2.** Marked up original gel electrophoresis presented in *Figure 1B* (panels 1 and 2).

**Source data 3.** Original gel electrophoresis presented in *Figure 1B* (panel 3).

**Source data 4.** Marked up original gel electrophoresis presented in *Figure 1B*.

**Source data 5.** Original immunoblot presented in *Figure 1C*.

**Source data 6.** Marked up original immunoblot presented in *Figure 1C*.

**Figure supplement 1.** Western blot of lysates prepared from mixed stage *Pf*3D7 wild type (WT) and *Pf*A-M17-HAglmS parasites probed with either pre-bleed rabbit serum or rabbit serum after multiple rounds of inoculation with *Pf*A-M17 recombinant protein (final bleed).

**Figure supplement 1—source data 1.** Original immunoblot presented in *Figure 1—figure supplement 1*.

**Figure supplement 1—source data 2.** Marked up original immunoblot presented in *Figure 1—figure supplement 1*.

parasites was obtained by limiting dilution. Diagnostic PCR confirmed that these parasites were positive for pM17-HAglmS integration (*Figure 1B*). Western blot analysis of whole parasite lysate confirmed expression of HA-tagged *Pf*A-M17, although the protein ran lower than its predicted molecular mass of 72 kDa (*Figure 1C*). This may be due to proteolytic cleavage during lysate preparation as *Pf*A-M17 harbors a low complexity region at its N-terminal end. Irrespective, lysate of wildtype *Pf*3D7 parasites run alongside *Pf*A-M17-HAglmS and probed with rabbit anti-M17 demonstrated a shift in molecular weight between the two proteins in accordance with the size of the epitope tags, which indicates that while the molecular mass of *Pf*A-M17 is smaller than expected, *Pf*A-M17 had been correctly tagged

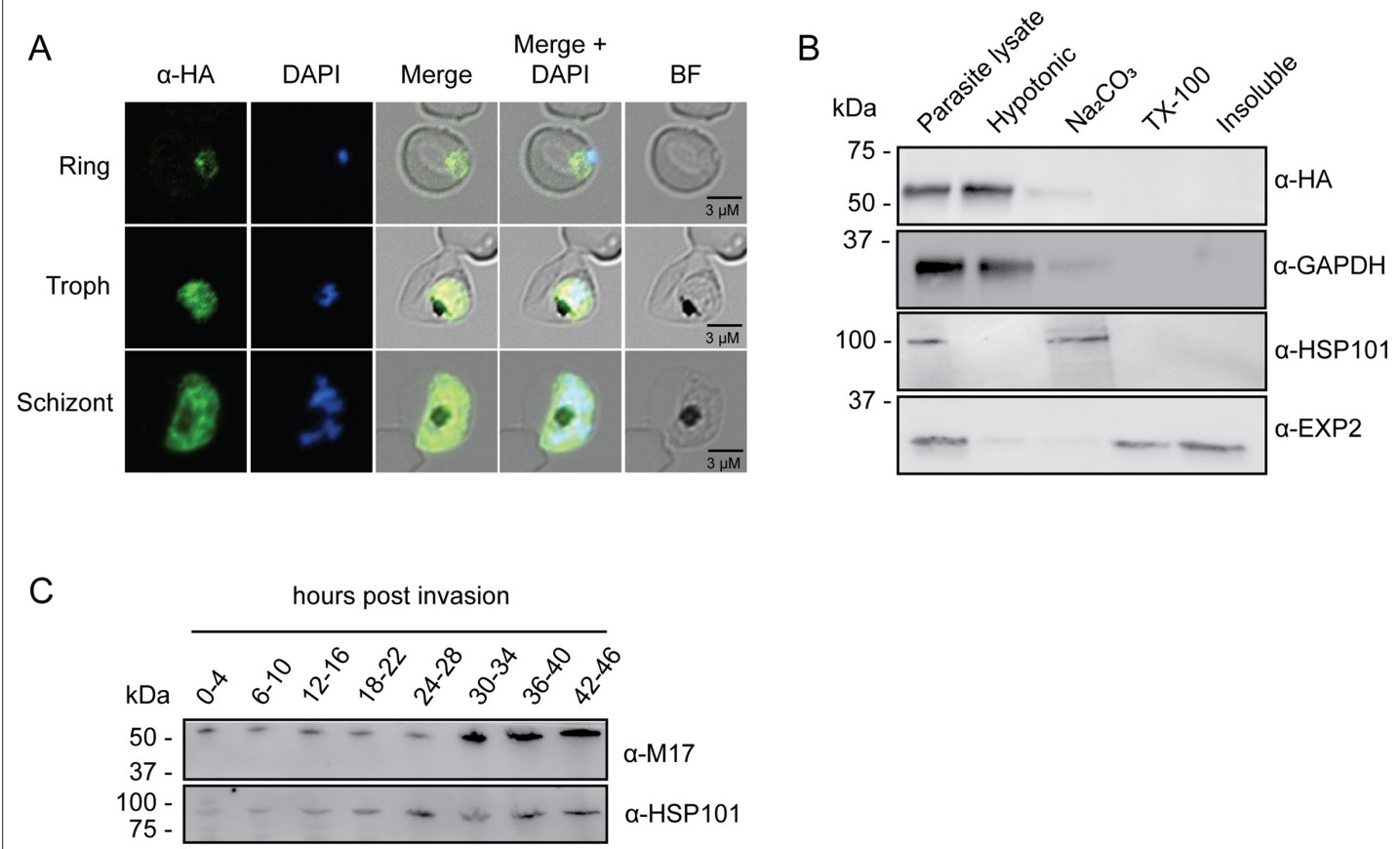

**Figure 2.** Analysis of *Pf*A-M17 localization and expression over the lifecycle. (**A**) Immunofluorescent analysis of *Pf*A-M17-HAglmS parasites in the three distinct lifecycle stages fixed with 90:10 acetone:methanol and probed with anti-HA and DAPI. (**B**) Saponin-lysed mixed stage *Pf*A-M17-HAglmS parasites were sequentially lysed in the buffers indicated from left to right and analyzed by Western blotting. Insoluble material represents the remaining pellet after lysis in 1% Triton X-100. GAPDH, HSP101, and EXP2 serve as controls for cytoplasmic, membrane-associated, and integral membrane proteins respectively. Blot is representative of three biological replicates. (**C**) Western blot analysis of endogenous *Pf*A-M17 expression in *Pf*3D7 wildtype parasites over the erythrocytic cycle probed with anti-M17 antibodies. HSP101 serves as a loading control.

The online version of this article includes the following source data for figure 2:

**Source data 1.** Original immunoblot presented in *Figure 2B* (anti-HSP101 blot).

**Source data 2.** Marked up original immunoblot presented in *Figure 2B* (anti-HSP101 blot).

**Source data 3.** Original immunoblot presented in *Figure 2B* (anti-GAPDH blot).

**Source data 4.** Marked up original immunoblot presented in *Figure 2B* (anti-GAPDH blot).

**Source data 5.** Original immunoblot presented in *Figure 2B* (anti-HA blot).

**Source data 6.** Marked up original immunoblot presented in *Figure 2B* (anti-HA blot).

**Source data 7.** Original immunoblot presented in *Figure 2B* (anti-EXP2 blot).

**Source data 8.** Marked up original immunoblot presented in *Figure 2B* (anti-EXP2 blot).

**Source data 9.** Original immunoblot presented in *Figure 2C*.

**Source data 10.** Marked up original immunoblot presented in *Figure 2C*.

with HA (*Figure 1—figure supplement 1*). Immunofluorescence analysis further confirmed HA expression, showing that *Pf*A-M17 was excluded from both the digestive vacuole and the parasite nucleus (*Figure 2A*). Sequential solubilization assays performed on mixed-stage parasite lysates showed that *Pf*A-M17 was released into the soluble fraction and was absent from membrane-associated or integral membrane fractions, supporting its cytosolic localization (*Figure 2B*). Western blot analysis of protein lysate harvested from *Pf*3D7 wild type parasites every 6 hr and probed with rabbit anti-M17 showed continuous expression throughout the asexual blood stages, with peak expression around 30 hr post invasion (hpi) as previously reported (*Figure 2C*; *Figure 1—figure supplement 1*).

## Knockdown of *Pf*A-M17 expression reveals its essentiality to parasite survival

The synthetic ribozyme incorporated into the 3'UTR of *Pf*A-M17-HAglmS parasites allows the knockdown of protein expression at the transcriptional level with addition of glucosamine (GlcN), allowing characterization of protein function and assessment of the proteins' contribution to parasite growth. Ring stage parasites at 0–4 hpi in cycle 1 were treated with 2.5 mM GlcN or left untreated. Parasites were harvested at trophozoite stage in cycle 1 and cycle 2 and significant protein knockdown was determined by Western blotting. This revealed that knockdown was efficient, with 84% and 92% knockdown in cycle 1 and 2, respectively (*Figure 3A*). Immunofluorescence analysis of parasites in cycle 2 following *Pf*A-M17 knockdown also revealed that parasites were no longer expressing the HA epitope tag, confirming loss of the protein (*Figure 3—figure supplement 1*). Parasite growth was determined by Giemsa-stained smears and compared to untreated parasites (*Figure 3B*). Whilst there was no delay in parasite growth in cycle 1, a significant delay in parasite growth was observed the cycle following knockdown (C2) (*Figure 3B and C*). This growth delay was already evident by early trophozoites stage and parasites reaching schizogony showed significant morphological changes. Few parasites went on to commence cycle 3, as evidenced by the significantly different parasitemias at 100 hr post-treatment (*Figure 3D*). Measurement of parasite survival after 10 days in culture was also assessed using a Sybr Green I assay, which revealed knockdown with GlcN was significantly detrimental to parasite growth (*Figure 3E*). None of these growth defects were evident in *Pf*3D7 parasites that had been treated with the same concentration of GlcN, confirming that the effect was due to the loss of *Pf*A-M17 (*Figure 3—figure supplement 2*). Overall, this demonstrates that loss of *Pf*A-M17 has a detrimental effect on parasite growth and indicates that this aminopeptidase is essential for survival of *P. falciparum*.

## Development of a *Pf*A-M17-specific inhibitor

Since knockdown of *Pf*A-M17 resulted in parasite death, we next sought to develop and characterize a specific inhibitor to target *Pf*A-M17 to use as a tool for functional studies. *Pf*A-M17 uses a metal dependent mechanism to hydrolyze the scissile peptide bond of peptide substrates, removing single amino acids from the N-terminal end of short peptides. Our parallel program of inhibitor design and synthesis identified a number of hydroxamic acid-based compounds that inhibit *Pf*A-M17 (*Drinkwater et al., 2016*; *Vinh et al., 2019*). In order to reduce polarity and improve water solubility, compound **3** (*Figure 4A*) was targeted. This compound possesses a 4-hydroxymethylphenyl group in place of the 3,4,5-trifluorophenyl group present in **6 l** (*Vinh et al., 2019*) which reduced the cLogP from 3.6 to 2.5.

Compound **3** was synthesized from the aryl bromide **1**, that we have reported previously (*Drinkwater et al., 2016*). The synthesis first involved the attachment of the 4-hydroxymethylphenyl moiety via a Suzuki coupling reaction to afford **2** (*Figure 4A*). The ester of **2** was subsequent converted to the corresponding hydroxamic acid via treatment with hydroxylamine hydrochloride under basic conditions (*Figure 4A*). Compound **3** was found to possess water solubility of >100 mg/mL as assessed by nephelometry using the methodology developed by *Bevan and Lloyd, 2000*. This was significantly higher than **6 l** from *Vinh et al., 2019* whose solubility was in the 12.5–25 mg/mL range.

The inhibitory activity toward purified, recombinant *Pf*A-M17 shows **3** to be a potent inhibitor ($K_i$ = 18 ± 3 nM) with excellent selectivity over *Pf*A-M1 ($K_i$ = 4424 ± 501 nM, *Figure 4—figure supplement 1*), which often shows cross-reactivity with *Pf*A-M17 substrates and inhibitors (*Drinkwater et al., 2016*; *Vinh et al., 2019*). To confirm the binding mode and suggest a reason for such excellent selectivity, we solved the 2.5 Å X-ray crystal structure of *Pf*A-M17 bound to **3**. The inhibitor binding modes were well conserved across all active sites and subsequently from here on in the molecular interactions

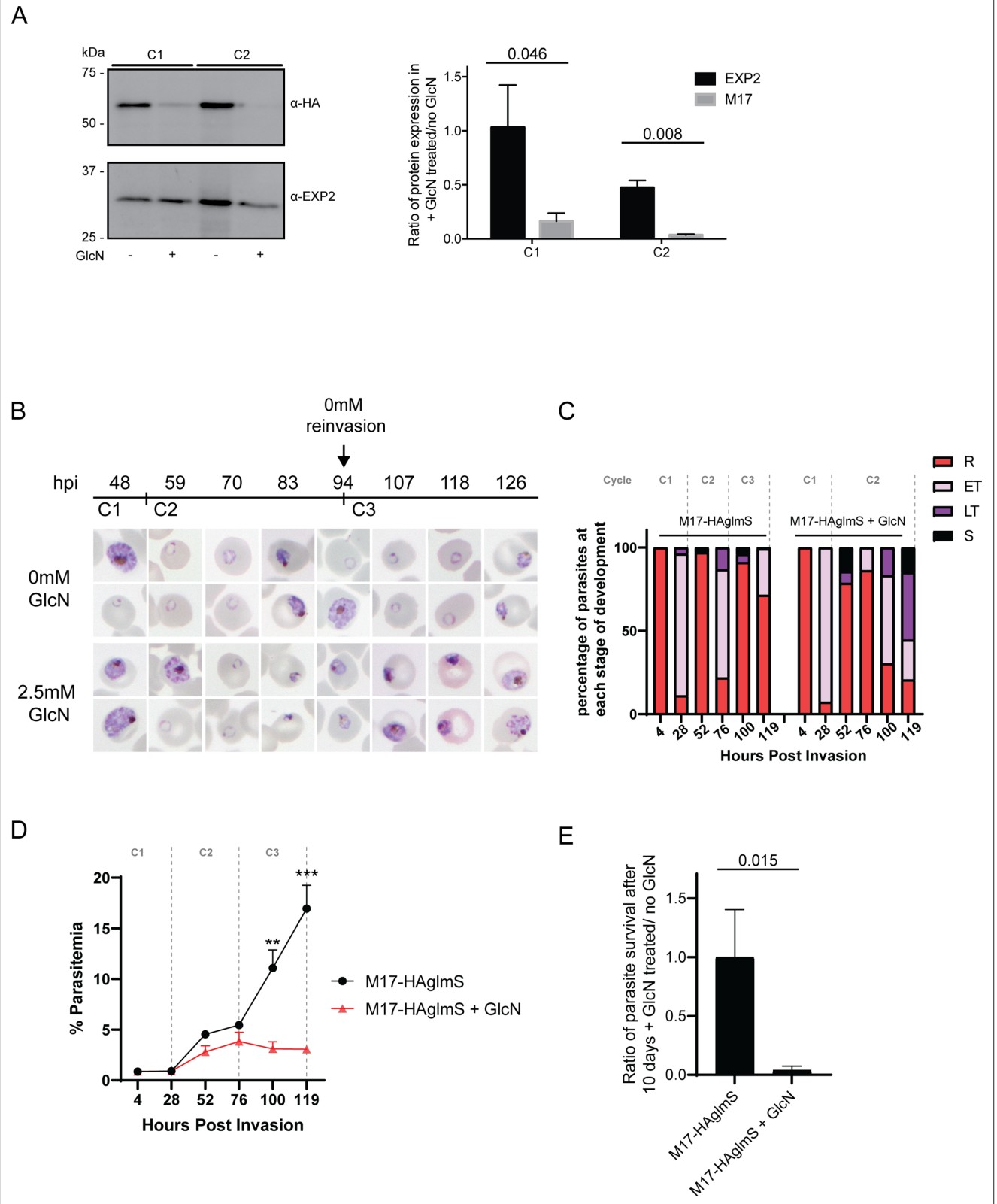

**Figure 3.** Depletion of *Pf*A-M17 expression leads to perturbed parasite growth *in vitro*. (**A**) Knockdown of *Pf*A-M17 expression. Left panel: Representative Western blot of *Pf*A-M17-HAglmS protein lysates prepared from parasites treated with either 2.5 mM GlcN (+) or left untreated (-) at trophozoite stage in cycle 1 (C1) or cycle 2 (C2). EXP2 serves as a loading control. Blot is representative of three biological replicates. Right panel: Densitometry of bands observed in western blots was performed using ImageJ to calculate the ratio of *Pf*A-M17 protein expression in GlcN-

*Figure 3 continued on next page*

*Figure 3 continued*

treated parasites relative to EXP2 compared to that of untreated parasites. Shown is the mean ± standard deviation (n=3). Statistical significance was determined using an unpaired t-test. (**B**) Representative Giemsa-stained parasite smears of *Pf*A-M17-HAglmS cultures treated with 0 mM or 2.5 mM GlcN shows depletion of *Pf*A-M17 protein results in a growth delay following reinvasion into cycle 2 (C2). (**C**) Percentage of *Pf*A-M17-HAglmS parasites at each stage of development ±GlcN over three cycles shows depletion of *Pf*A-M17 leads to delayed parasite development within cycle 2 (n=3 biological replicates) (**D**) Parasitemias of *Pf*A-M17-HAglmS parasites cultured ±GlcN over three cycles. Invasion into cycle 2 is not significantly affected but the growth delay of GlcN-treated parasites observed in this cycle affects parasitemia thereafter, with a significant difference in parasitemia by the time untreated *Pf*A-M17-HAglms have entered cycle 3. Grey dotted lines are representative of the time when untreated parasites have completed a cycle. Plotted is the mean ± standard deviation (n=3), with statistical significance determined using an unpaired t-test (**p≤0.01, ***p≤0.001). (**E**) Ratio of parasite survival of *Pf*A-M17-HAglmS after treatment with GlcN for 10 days compared to untreated parasites as determined by Sybr Green 1 assay. Shown is the mean ± standard deviation (n=3). Statistical significance was determined using an unpaired t-test.

The online version of this article includes the following source data and figure supplement(s) for figure 3:

**Source data 1.** Original immunoblot presented in *Figure 3A*.

**Source data 2.** Marked up original immunoblot presented in *Figure 3A*.

**Figure supplement 1.** Immunofluorescent analysis of *Pf*A-M17-HAglmS parasites the cycle following the addition of glucosamine (GlcN; C2) fixed with 90:10 acetone:methanol and probed with anti-HA, anti-EXP2 and DAPI.

**Figure supplement 2.** Addition of Glucosamine does not significantly affect growth of *Pf*3D7 parasites.

of **3** with the active site is described only for Chain A. Analysis of the X-ray crystal structure of *Pf*A-M17 bound to **3** showed that the position and orientation of the compound was similar to our previous inhibitors with the hydroxamic acid core coordinating to the zinc ions in the active site (*Figure 4B*, *Figure 4—figure supplement 1*). The two zinc ions are coordinated by the hydroxamic oxygens, that also form hydrogen bonds with the conserved carbonate ion and catalytic residue Lys386 (*Figure 4—figure supplement 1*). In the S1 pocket, the *tert*-butyl makes no interactions with the body of the protein, but the amide carbonyl of **3** forms a hydrogen bond with the main chain amide of Gly489 as well as a water molecule (*Figure 4—figure supplement 1*). The 4-hydroxymethylphenyl group that replaced the trifluorophenyl group present in **6** f (*Vinh et al., 2019*) packed with the hydrophobic residues Leu487, Gly489, Leu492 Met396, Phe583, and Ala577 and the hydroxyl group of **3** can interact with the sulfur atom of Met392 (*Figure 4B*, *Figure 4—figure supplement 1*).

To try and understand why **3** could act as a selective inhibitor of *Pf*A-M17 and showed very little activity toward *Pf*A-M1, we attempted to solve the X-ray crystal structure of *Pf*A-M1 bound to **3**. This was unsuccessful and no compound density was observed within any datasets collected from co-crystalized or soaked *Pf*A-M1 crystals. This was not surprising as our previous attempts to collect structures of *Pf*A-M1 crystals bound with weak inhibitors have always failed. Using an existing structure as a template (4ZX4.pdb) of *Pf*A-M1 bound to a hydroxamic acid inhibitor that possessed a 3,4,5-trifluorophenyl group rather than the 4-hydroxymethylphenyl found in **3**, we were able to superpose **3** onto the 4ZX4.pdb ligand however inspection of the S1 pocket with the superposed ligand does not identify why this compound shows low potency toward *Pf*A-M1. Replacement of the trifluorophenyl group for 4-hydroxymethylphenyl positioned the hydroxyl group of **3** close to E572 in the S1 pocket (~2.8 Å from the Cα atom and 1.7 Å from the Cγ atom). However, it would be surprising if this close contact was the reason for the lack of potency for this inhibitor in that E572 has been shown to move to accommodate bulky hydrophobic groups that extend from longer inhibitors (*Velmourougane et al., 2011*) and the hydroxyl group and the E572 side-chain would likely be able to re-position/rotate to avoid a clash. In the S1' pocket, the pivalamide moiety is easily positioned to the same place as other compounds with a similar scaffold but also does not suggest an obvious reason for the lack of potency with regard to the fit of the compound into the active site or its substrate pockets. It may be that the change in electronegativity, via the loss of the 3 fluorine atoms, results in the compound not being able to access the buried active site with the same affinity.

## The *Pf*A-M17-specific inhibitor 3 kills parasites within a sub-micromolar range

With the inhibitor in hand, its effectiveness on parasites was next tested. The $EC_{50}$ of **3** on *Pf*3D7 was determined to be 326 nM (145–581 CI) using a standard 72 hr ring killing assay, whereby parasite survival was determined over a range of compound concentrations (*Figure 4C*). To determine at what stage of the erythrocytic cycle **3** impacts on parasite growth, parasite cultures were treated with

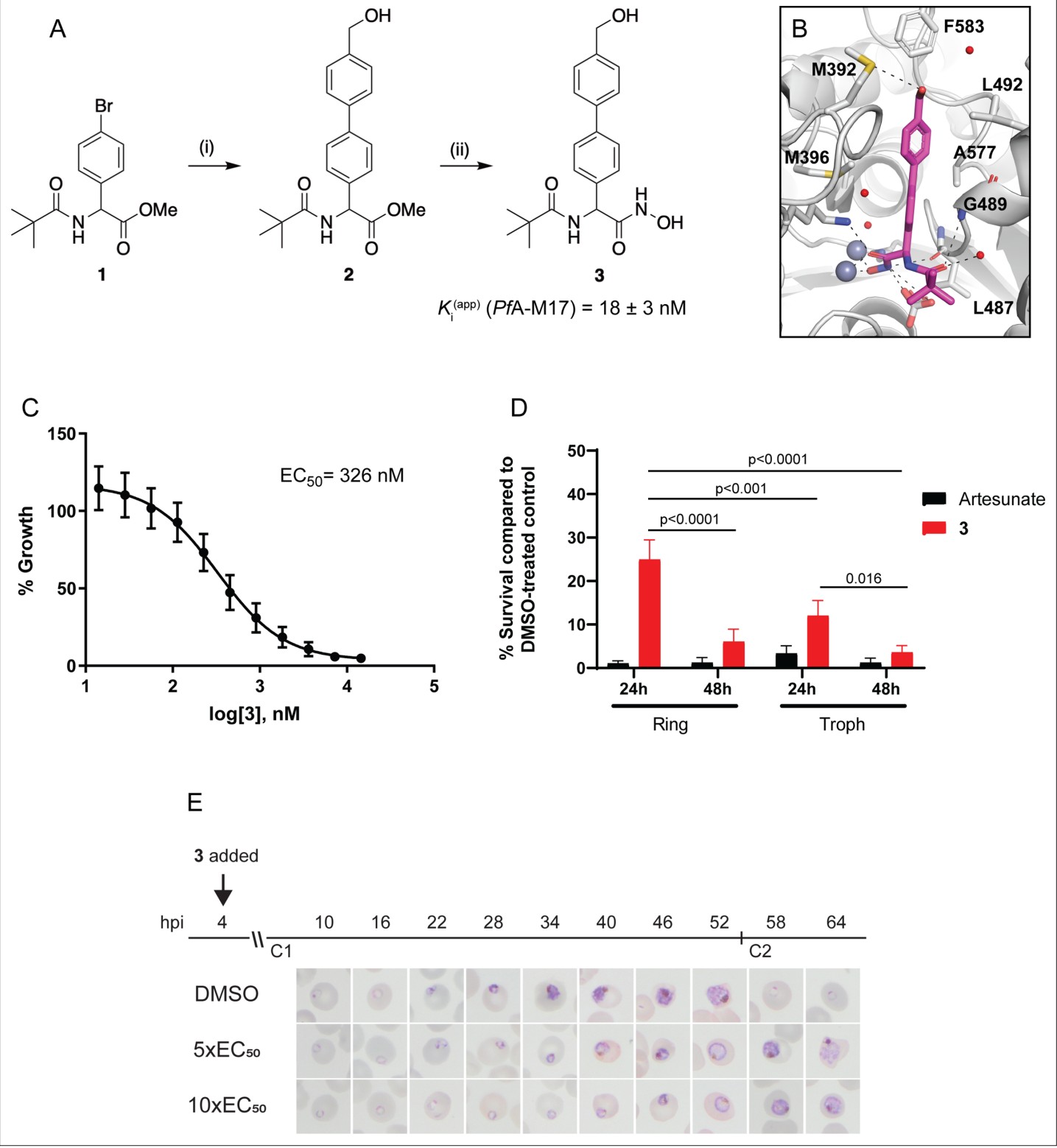

**Figure 4.** Synthesis and activity of **3,** a specific *Pf*A-M17 inhibitor. (**A**) Scheme 1. Synthesis of **3**: (i) Boronic acid, Pd(PPh$_3$)$_2$Cl$_2$, Na$_2$CO$_3$, THF, 100 °C, 2 hr, (ii) NH$_2$OH.HCl, KOH, RT, 16 hr. Inhibition constant for **3** toward recombinant, purified *Pf*A-M17 is shown. (**B**) Binding mode of **3** bound to *Pf*A-M17. Solvent-accessible surface of *Pf*A-M17 (grey) with active site ions shown in grey spheres. Stick representation (magenta) shows the binding positions of **3**. Molecular interactions between **3** and *Pf*A-M17 are indicated by dashed lines; water molecules are represented by red spheres. (**C**) Killing action of **3** over 72 hr as determined by SYBR Green I assay. The EC$_{50}$ value was calculated from four biological replicates performed in triplicate and data plotted

*Figure 4 continued on next page*

*Figure 4 continued*

as the mean ± standard error of the mean. (**D**) Parasite killing rate was determined by incubating *Pf*3D7 parasites in 10 x $EC_{50}$ as previously determined for either 24 or 48 hr before the drug was washed off and parasites allowed to grow for a further 48 hr. Survival was determined via Sybr Green I assay and compared to vehicle (DMSO)-treated controls. Shown is the mean ± standard deviation (n=4). Statistical significance was determined using a one-way ANOVA. (**E**) Synchronized parasites at 4 hr post-invasion (hpi) were treated over two cycles (C1, cycle 1; C2, cycle 2) with either 5 x or 10 x $EC_{50}$ or DMSO at the concentration present in the 10 x $EC_{50}$ treatment. Representative Giemsa-stained smears from two biological replicates show delay in parasite maturation to schizogony (5 x $EC_{50}$) or trophozoite stage (10 x $EC_{50}$).

The online version of this article includes the following figure supplement(s) for figure 4:

**Figure supplement 1.** Compound **3** is a potent and selective *Pf*A-M17 inhibitor.

either 10 x $EC_{50}$ **3** or artesunate ($EC_{50}$ 4.0 nM, 0.5–6.5 CI) for 24 or 48 hr, commencing at either the ring or trophozoite stage, and following compound washout, cultures were incubated for a further 48 hr (*Figure 4D*). This revealed that **3** was less effective at killing parasites with only a 24 hr treatment period compared to the 48 hr treatment, irrespective of whether the compound was administered at ring or trophozoite stage. Treatment for only 24 hr was significantly more effective when added at the trophozoite stage compared to ring stage addition. The most effective killing for **3** was observed when trophozoite stage parasites were treated for 48 hr. These findings were consistent with the observation that expression of *Pf*A-M17 peaks within the rapid trophozoite growth stage, with much lower expression during the ring stage. Moreover, it also corresponds with the growth delay observed at trophozoite stage after knockdown of *Pf*A-M17 protein expression. Analysis of Giemsa-stained *Pf*3D7 parasites treated with 5 x or 10 x the $EC_{50}$ showed that both treatments resulted in a growth delay commencing around early trophozoite stage when compared to the vehicle control DMSO-treated parasites and parasites had either not entered schizogony (10 x $EC_{50}$ treatment) or only just completed schizogony (5 x $EC_{50}$ treatment) 64 hr later (*Figure 4E*).

## Analysis of parasites depleted of *Pf*A-M17 alongside treatment with the specific *Pf*A-M17 inhibitor

To determine the specificity of **3** for *Pf*A-M17 and whether it may exhibit off-target effects, we initially attempted to generate drug resistant parasites using treatment with 3 x $EC_{90}$ as previously described (*Duffey et al., 2021*); however, no parasites were able to be recovered from five independent replicates. Next, we examined the sensitivity of parasites depleted of *Pf*A-M17 to **3**. For these experiments, *Pf*A-M17-HAglmS parasites depleted of *Pf*A-M17 using 2.5 mM GlcN were treated with 5 x or 10 x the $EC_{50}$ of **3** the cycle following knockdown just after reinvasion and before any death is occurring due to the knockdown (*Figure 3B*) and the growth normalized to *Pf*A-M17 expressing parasites (i.e. not exposed to GlcN treatment) treated with DMSO or 5 x the $EC_{50}$. Analysis of growth using Sybr Green I assay showed that there was no significant difference in growth curves between parasites depleted of *Pf*A-M17 and those additionally treated with **3** (*Figure 5A*). The reduction in growth at 94 hpi of all treated lines is consistent with the observation that parasites fail to reinvade and commence a subsequent cycle, unlike parasites treated with DMSO alone (*Figure 5A*). Moreover, analysis of Giemsa-stained PfA-M17-HAglmS parasites showed that the addition of 5 x or 10 x the $EC_{50}$ of **3** in the cycle following depletion of *Pf*A-M17 showed a comparable phenotype to *Pf*A-M17 knockdown parasites treated with DMSO (*Figure 5B*). These results are in keeping with **3** not having any significant off-target effects and that the effect of this compound on parasites could be through inhibition of *Pf*A-M17.

## Thermal proteomics profiling (TPP) confirms *Pf*A-M17 is the target of compound 3

As an alternative approach to identify the potential target(s) of **3** given the inability to generate resistant parasites for whole genome sequencing, TPP was performed. This methodology allows unbiased identification of the binding target in the *P. falciparum* proteome. A single TPP experiment with four technical replicates (i.e. four independent incubations of **3** with protein lysate) and a thermal challenge of a single temperature of 60°C with low (3 μM (10 x $EC_{50}$)), and high (12 μM (40 x $EC_{50}$)) compound **3** concentrations was used to identify a concentration dependent stabilization of the target protein. Following the thermal challenge, proteins were centrifuged to remove denatured protein that had precipitated,

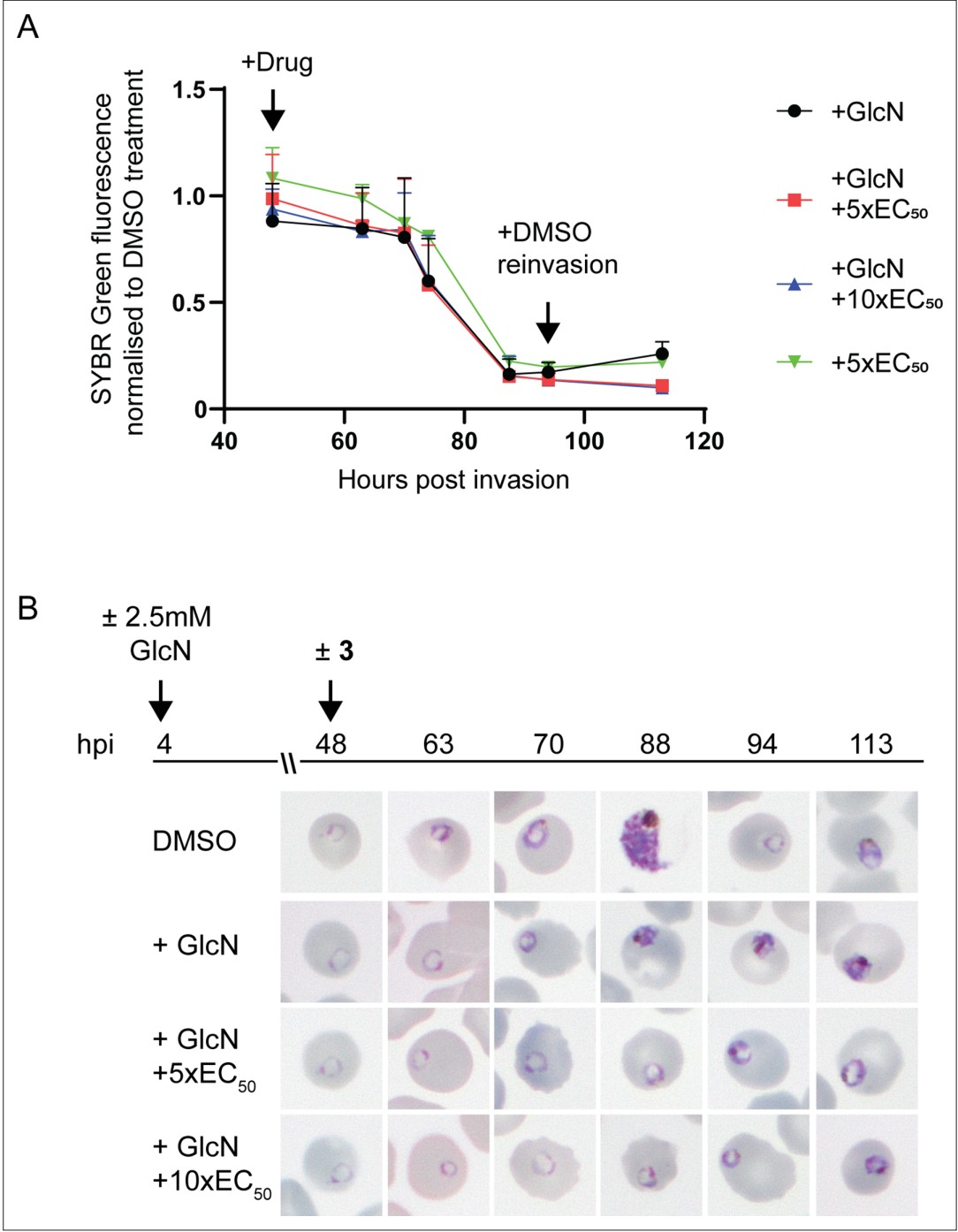

**Figure 5.** Effect of **3** treatment in parasites depleted of *Pf*A-M17. (**A**) SYBR I Green fluorescence normalized to DMSO treated parasites indicates there is no significant difference on growth between treatments. Plotted is the mean ± standard deviation (n≥2). (**B**) Upper panel: overview of experiment. Heparin synchronized *Pf*A-M17-HAglmS parasites were treated with 2.5 mM GlcN and allowed to invade into cycle 2 before being treated with **3** or DMSO. Lower panel: representative Giemsa-stained parasite smears of *Pf*A-M17-HAglmS parasites treated with ±GlcN and ± 5 x or 10 x EC50 of compound **3** shows parasites developing at a slower rate than the DMSO-treated control but in a similar manner between the treatment groups, n=2 biological replicates.

with the expectation that any protein stabilized by compound **3** would be detected in higher abundance in the remaining solution. 1883 proteins were identified by LC-MS and only a single protein had an altered thermal profile (fold change >1.15 and p-value <0.05) across the two concentrations of compound **3** (3 µM and 12 µM) compared to the DMSO control (*Figure 6A*). This protein was *Pf*A-M17 (Pf3D7_1446200), which demonstrated a 1.15-fold change stabilization following incubation with 3 µM and 1.26-fold change stabilization following incubation with 12 µM of compound **3,** compared to 0 µM control (*Figure 6B*). Furthermore, the TPP data demonstrated that *Pf*A-M1 was not the target, as its thermal profile following compound **3** addition was unchanged. Overall, using this unbiased thermal stabilization proteomics approach, we have shown that compound **3** is selectively targeting *Pf*A-M17.

## Loss of *Pf*A-M17 results in the formation of multiple digestive vacuoles

Analysis of Giemsa-stained smears of *Pf*3D7 parasites treated with **3** and *Pf*A-M17-HAglmS parasites in cycle 2 following knockdown revealed that some parasites at early trophozoite harbored multiple hemozoin (Hz) crystals. To determine the significance of this finding, *Pf*A-M17-HAglmS Giemsa stained parasites, alongside *Pf*3D7 parasites, were scored for their number of individual hemozoin crystals under light microscopy the cycle after GlcN addition; only singly infected RBCs were counted, and the rest excluded (*Figure 7A*). *Pf*3D7 parasites treated with 5 x and 10 x the $EC_{50}$ of **3** or with the DMSO vehicle control were also scored in the same manner. *Pf*A-M17-HAglmS knockdown parasites had significantly more Hz crystals per parasite compared to untreated parasites as determined by a non-parametric Dunn's post-hoc test (*Figure 7A*), with approx. 30% of parasites containing multiple Hz crystals (all controls <10%). The addition of GlcN to *Pf*3D7 parasites did not have a significant effect on Hz crystal numbers, indicating that this phenotype was not the result of treatment with GlcN but specific to the loss of *Pf*A-M17. Similar results were seen in **3** treated parasites, with approx. 40% of parasites containing multiple Hz crystals per parasite for both the 5 x and 10 x $EC_{50}$ treatments when compared to the vehicle control (8.5%) (*Figure 7B*).

To determine if these Hz crystals were representative of multiple DVs or loss of DV integrity, transmission electron microscopy (TEM) was used to image parasites treated with **3**. Synchronised *Pf*3D7 parasites were treated with 10 x $EC_{50}$ and allowed to mature to early trophozoite stage before being processed for TEM; DMSO-treated parasites were also taken at a similar developmental stage and used as a control. Control parasites generally contained one large DV that was membrane bound and contained Hz crystals (white block-shaped; *Figure 7C*, upper panel), while the cytoplasm contained numerous free ribosomes (electron dense particles; *Figure 7C*, upper panel). In comparison, many parasites treated with **3** contained multiple profiles of DVs, each membrane bound and containing Hz crystals in a similar arrangement to that seen in the control (*Figure 7C*, lower panel). Some of the treated parasites also exhibited separation of membranes at the parasite periphery, a phenotype previously seen after treatment with antimalarials that could be due to drug mode of action, loss of parasite integrity or shrinkage of parasites during processing (*Sachanonta et al., 2011*). While some parasites treated with **3** still only contained one DV, as was also seen under light microscopy, no parasites imaged in the control group contained more than one DV profile, indicating that the development of multiple DVs is specific to the loss of *Pf*A-M17.

As the multiple DVs in parasites depleted of *Pf*A-M17 often contained Hz, this indicated that parasites were still capable of digesting hemoglobin. We next determined if the increase in number of DVs resulted in an overall increase in hemozoin formation, and thus hemoglobin digestion. Accordingly, *Pf*A-M17-HAglmS and *Pf*3D7 parasites were harvested at developmentally similar early trophozoite stages the cycle following GlcN addition and the quantity of hemozoin, as free heme, was measured. Between all groups, there was no statistical difference after the addition of GlcN, and while there appeared to be an upward trend in the free heme in GlcN treated *Pf*A-M17-HAglmS parasites compared to untreated, this did not reach statistical significance (*Figure 7D*). Thus, while the loss of *Pf*A-M17 results in the development of multiple DVs, it does not increase the quantity of free heme, representative of hemoglobin digestion and hemozoin production, suggesting that there is not an upstream effect on this digestive pathway.

## *Pf*A-M17 plays a role in the degradation of hemoglobin-derived peptides

To further examine the specificity of **3** and to determine why knockdown of *Pf*A-M17 has a drastic effect on parasite growth, we compared the metabolomic profiles of *Pf*A-M17 parasites and *Pf*3D7

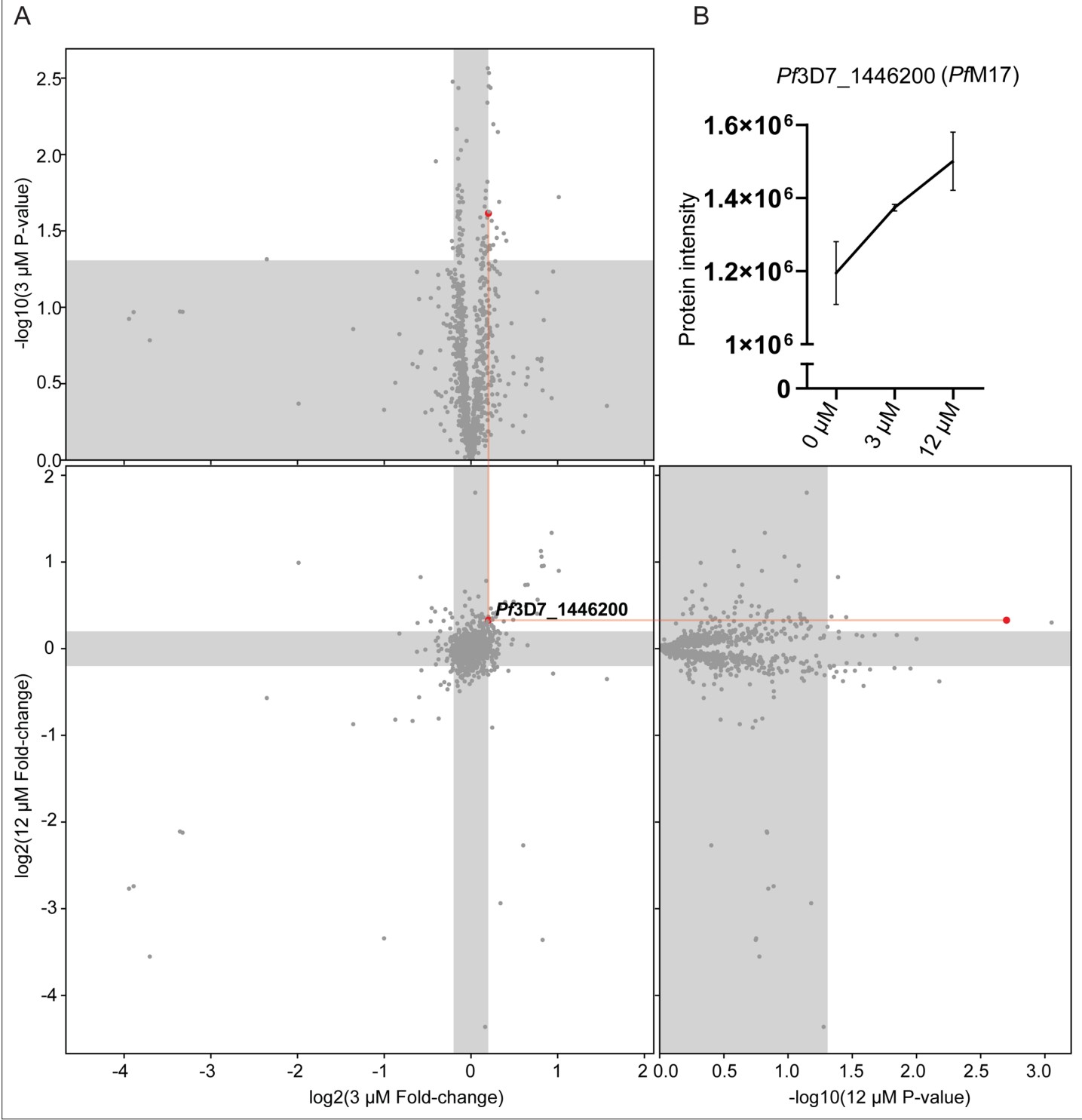

**Figure 6.** Thermal proteome profiling identified *Pf*A-M17 to be the target of compound **3**. (**A**) Paired volcano plot of all proteins detected. The outside panels show the log2 fold change vs –log10 p-value of proteins following treatment with 3 µM or 12 µM of compound **3**, relative to the 0 µM negative control, following a 60 °C thermal challenge. Proteins significantly (p-value > -log10 (0.05) Welch's t test) stabilized (fold change >log2 (1.15)) or destabilized (fold change <log2 (0.87)) appear in the unshaded regions. The thermal stability of a single protein (*Pf*3D7_1446200) was altered at both concentrations with a p- value <0.05, with increasing stability in increasing concentrations of compound **3**. This is indicated via red lines and dots. (**B**) Protein intensity of *Pf*3D7_1446200 (*Pf*A-M17) in increasing concentration of compound **3** and following 60 °C thermal challenge. Value represents the mean of four technical replicates ± standard deviation.

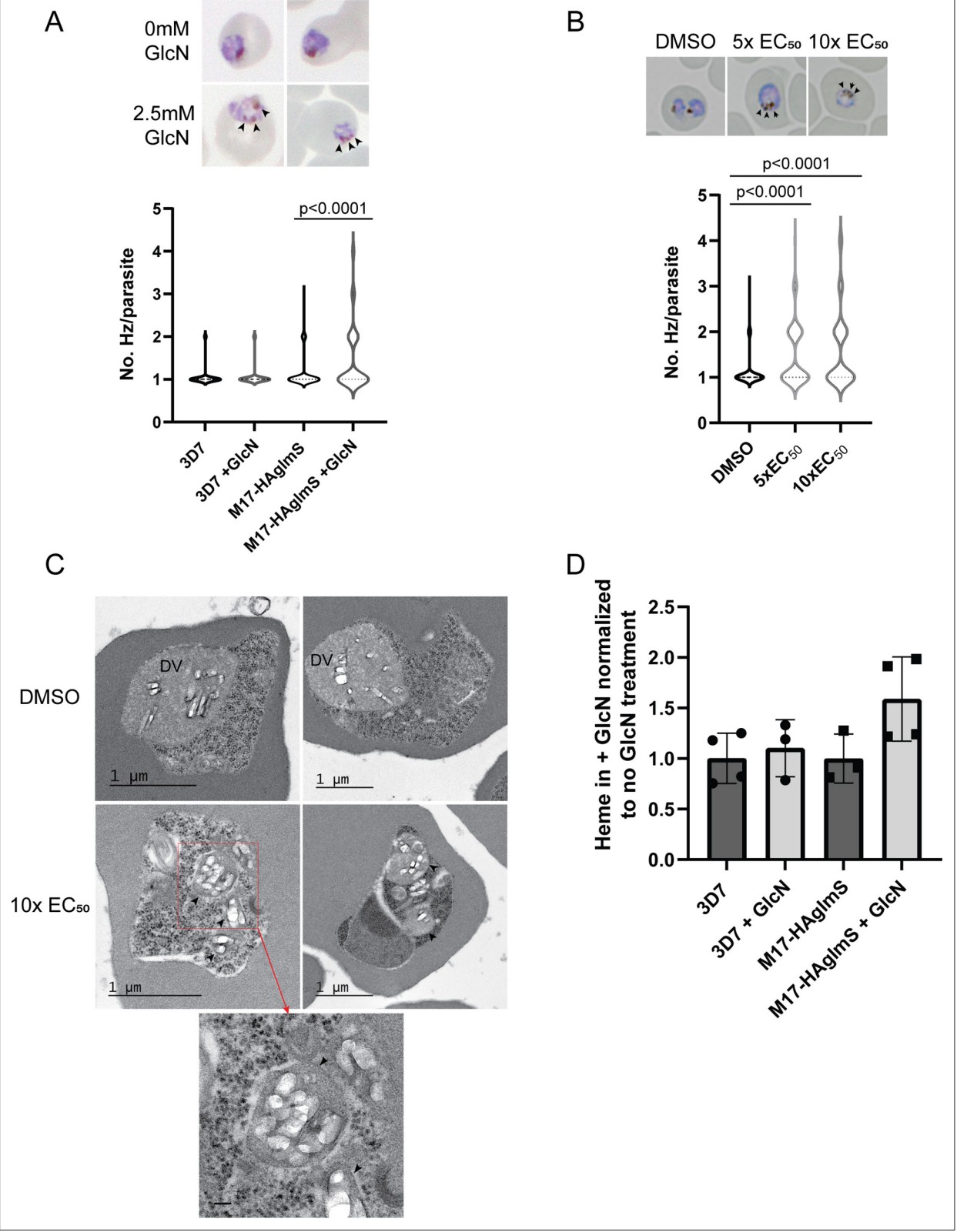

**Figure 7.** Parasites depleted of *Pf*A-M17 develop significantly more digestive vacuoles. (**A**) Upper panel: Representative Giemsa-stained smears of *Pf*A-M17-HAglmS ± GlcN; black arrowheads indicate multiple digestive vacuoles. Lower panel: Number of digestive vacuoles per *Pf*3D7 and *Pf*A-M17-HAglmS the cycle following addition of glucosamine (GlcN) as determined under Giemsa-staining. Shown is the median with range combined from four biological replicates (n≥100). Statistical significance was determined by a one-way ANOVA followed by Dunn's Multiple Comparison test (a

*Figure 7 continued on next page*

*Figure 7 continued*

nonparametric post hoc). (**B**) Upper panel: Representative Giemsa-stained smears of *Pf*3D7+ DMSO, 5 x or 10 x the $EC_{50}$ of **3**; black arrows indicate multiple digestive vacuoles. Lower panel: Number of digestive vacuoles per *Pf*3D7 following the addition of **3** at 4 hr post invasion (hpi) as determined under Giemsa-staining. Shown is the median with range combined from two biological replicates (n≥100). Statistical significance was determined by a one-way ANOVA followed by Dunn's Multiple Comparison test (a nonparametric post hoc). (**C**) Representative transmission electron micrographs of two *Pf*3D7 parasites treated with 10 x $EC_{50}$ of **3** (lower panels) and two with DMSO vehicle control (upper panels). Images show hemozoin crystals (white block shapes) bound within digestive vacuoles (DVs). DVs in treated parasites are indicated by black arrowheads. In the higher magnification micrograph of the lower left panel (corresponding area indicated by red dashed box), the arrowheads indicate the membrane surrounding each of the DV profiles. Scale bar in the high magnification represents 100 nm. (**D**) Free heme representative of hemozoin in *Pf*3D7 and *Pf*A-M17-HAglmS GlcN-treated parasites relative to untreated parasites. Shown is the mean ± standard deviation (n≥3). No significance difference between groups was found using an unpaired t-test.

parasites grown in the presence and absence of GlcN and *Pf*3D7 parasites treated with 10 x the $EC_{50}$ of **3** for 1 hr. Principle component analysis and heatmap analysis of relative abundances of putative metabolites dysregulated following *Pf*A-M17 knockdown and parasites treated with **3** shared a common, prominent metabolic signature: the increase in a series of peptides (***Figure 8***). This was then confirmed by two independent experiments, experiment 2 (***Figure 8—figure supplement 1***), which only analyzed parasites in which *Pf*A-M17 expression had been depleted, and experiment 3 (***Figure 8—figure supplement 2***), which only analyzed parasites treated with **3**. Targeted analysis of the common set of 149 peptides identified in all three experiments demonstrated that all of the 77 peptides that were significantly (p-value <0.05) increased in abundance in parasites depleted of *Pf*A-M17 were also significantly elevated following treatment with **3** (***Figure 8—figure supplement 3***). Likewise, the vast majority of the 80 significantly increased peptides following treatment with **3** also displayed significantly elevated abundance (fold change >1.5) following *Pf*A-M17 depletion, with the exception of three peptides (Lys-Gly, Glu-Glu-Glu-Lys-Trp, and Asp-Phe-Ile-Tyr-Tyr) that were only enriched following treatment with **3** (***Figure 9A***). The 80 peptides identified were then analyzed to determine whether they may be derived from hemoglobin (***Florens et al., 2002***). First, we used MS/MS spectra to confirm the sequence of nearly half of the peptides and assessed whether these sequences could be mapped to one of the hemoglobin chains α, β, or δ (***Figure 9B***; yellow dots). For the remaining peptides for which MS/MS spectra could not be obtained, we assessed whether any peptide isomeric to the peptide identified by accurate mass could be mapped to hemoglobin (***Figure 9B***; blue dots). Overall, ~82% of significantly dysregulated peptides could be mapped to hemoglobin, and these peptides tend to increase in abundance on *Pf*A-M17 depletion or inhibition with **3** considerably more than peptides that cannot be hemoglobin-derived.

Although this result is consistent with the dysregulated peptides being predominantly hemoglobin-derived, the statistical significance of the effect is difficult to assess. To this end, we repeated this analysis for each of the ~4700 proteins identified in our recent comprehensive proteomic analysis of *P. falciparum*-infected erythrocytes and quantified the number of peptide matches to each protein and then divided by protein length to yield a normalized estimate of the similarity of each protein to our significantly dysregulated peptides (***Siddiqui et al., 2022***). By this measure, hemoglobin chains α, α2, β, and δ are four of the five most-similar proteins (***Figure 9C***). The remaining highly matched protein is human MYL4, which contains a repetitive proline-rich region resulting in multiple matches to short proline-containing peptides which are abundant in our dataset, so it's score may be artificially inflated by this sequence repetition. In this sense, peptides perturbed by *Pf*A-M17 disruption are significantly more 'hemoglobin-like' than the other proteins present in infected erythrocytes. Thus, metabolomics analysis of parasites depleted of *Pf*A-M17 or treated with **3**, suggests that *Pf*A-M17 is predominantly, but not exclusively, involved in hemoglobin digestion.

## Parasites grown in amino-acid-free medium containing only isoleucine become sensitized to the *Pf*A-M17 inhibitor 3

Since a function of hemoglobin digestion is to provide free amino acids to the parasite, we next determined if removal of exogenous amino acids, with the exception of isoleucine, from the culture medium would sensitize parasites to **3**. A standard 72 hr killing assay was used to determine the $EC_{50}$ of the compound on *Pf*3D7 parasites cultured concurrently in normal medium containing all amino acids, or amino-acid-free medium containing only isoleucine. This showed that parasites became

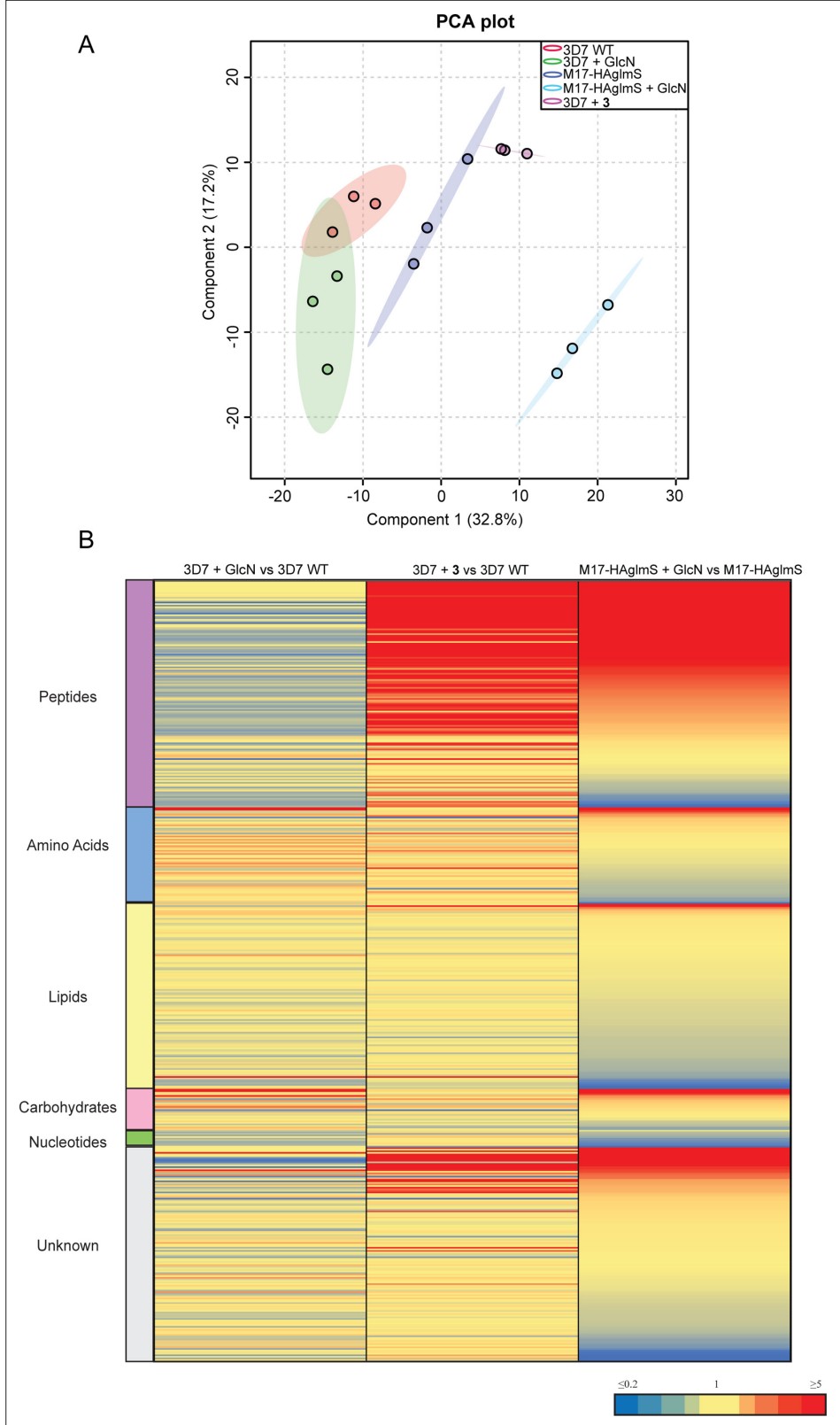

**Figure 8.** Untargeted metabolomics analysis of *Pf*A-M17-HAglmS and *Pf*3D7 parasites treated with ± GlcN and of *Pf*3D7 parasites treated with **3** from experiment 1. (**A**) Principal component analysis (PCA) of parasites (*Pf*A-M17-HAglmS and *Pf*3D7) treated with ± GlcN and **3** or DMSO control. Scores plot show principal components one and two, data points indicate individual sample replicates within each condition and the shaded area denotes

*Figure 8 continued on next page*

*Figure 8 continued*

95% confidence interval. (**B**) Heatmap showing the average fold change for all putative metabolites for the three treatment conditions of *Pf*3D7+ GlcN versus WT, *Pf*3D7 +compound **3** versus WT, and *Pf*A-M17HAglmS + GlcN versus *Pf*A-M17HAglmS. For *Pf*A-M17HAglmS + GlcN versus *Pf*A-M17HAglmS, the fold change values have been ordered from highest to lowest. Values represent the average of three technical replicates, red, blue, and yellow indicates increase, decrease and no change, respectively, in the fold change of putative metabolites identified.

The online version of this article includes the following figure supplement(s) for figure 8:

**Figure supplement 1.** Untargeted metabolomics analysis of *Pf*A-M17-HAglmS and *Pf*3D7 parasites treated with ±GlcN from experiment 2.

**Figure supplement 2.** Untargeted metabolomics analysis of *Pf*3D7 parasites treated with **3** and DMSO control from experiment 3.

**Figure supplement 3.** Targeted analysis of all common peptides identified from experiment 1, 2 and 3.

significantly more sensitive to **3** in amino-acid-free media, indicating that its target *Pf*A-M17 is responsible for supplying parasites with amino acids essential for survival (***Figure 10***). Comparatively, the loss of exogenous amino acids did not significantly impact the $EC_{50}$ of artemisinin as has previously been shown (***Harbut et al., 2011***), with artemisinin being a ring-stage killer that does not directly target hemoglobin digestion. That parasites are not sensitized to artemisinin after the removal of exogenous amino acids appears to be assay-dependent (***Giannangelo et al., 2020***).

## Discussion

For many years, *Pf*A-M17 has been implicated in the final stage of hemoglobin digestion without any definitive confirmation. Here, using a well-established conditional system to knockdown expression of *Pf*A-M17, we demonstrate that specific loss of this protein leads to delayed growth and eventual stalling of parasite development upon transition into trophozoite stage in the cycle following knockdown of the protein. This delay in parasite death is likely attributed to the level of knockdown occurring within the first cycle. The resulting phenotype is fatal, with parasites unable to propagate and advance through further cycles, consistent with previous conclusions that *Pf*A-M17 is likely to be an essential protein on the basis that *Pfa-m17* is refractory to gene knockout (***Dalal and Klemba, 2007***; ***Zhang et al., 2018***). This validates *Pf*A-M17 as a potential novel drug target.

Unlike previous studies on *Pf*A-M17, which have used external activity-based probes or compounds, the conditional knockdown strategy negates any off-target effects that may have convoluted the dissection of its function. Using the approaches herein, we were additionally able to show that **3,** specifically designed to target *Pf*A-M17, was on-target using unbiased thermal proteomics profiling, and that parasites treated with this compound displayed a comparable phenotype and metabolic profile to parasites in which *Pf*A-M17 expression had been conditionally depleted. We were unable to generate parasites resistant to **3**, which is not necessarily surprising given *Pf*A-M17 is an essential protein, and it may be impervious to mutations that permit parasite survival under the drug doses used here. Wash out experiments showed that parasites were more susceptible to **3** after they had been treated during at least one trophozoite phase, keeping with *Pf*A-M17 appearing to be essential for this stage of growth as seen with the knockdown phenotype. Analysis of the significantly disrupted metabolites after *Pf*A-M17 knockdown or treatment with **3** revealed an increase in abundance of peptides, many of which are likely to be derived from hemoglobin as MS/MS analysis shows the sequences of these peptides to be more similar to hemoglobin than other host or parasite proteins. This suggests that the failure to generate a sufficient pool of some amino acids from hemoglobin through a reduction in aminopeptidase activity may be the cause of parasite death. That *P. falciparum* parasites were significantly more sensitized to **3** when cultured in the absence of exogenous amino acids also suggests that the main function of *Pf*A-M17 is to provide amino acids for parasite growth.

***Harbut et al., 2011*** previously showed that *P. falciparum* cultured in the presence of an activity-based probe designed to inhibit *Pf*A-M17 resulted in parasite death at the early ring stage of the parasite lifecycle, suggesting that the role of *Pf*A-M17 in hemoglobin digestion may be subservient to an additional but essential role it plays earlier in the lifecycle (***Harbut et al., 2011***). The authors hypothesized this additional role is in the turnover of peptides originating from the proteosome, as proteasome inhibitors display ring-stage killing (***Prasad et al., 2013***). While we cannot rule out the

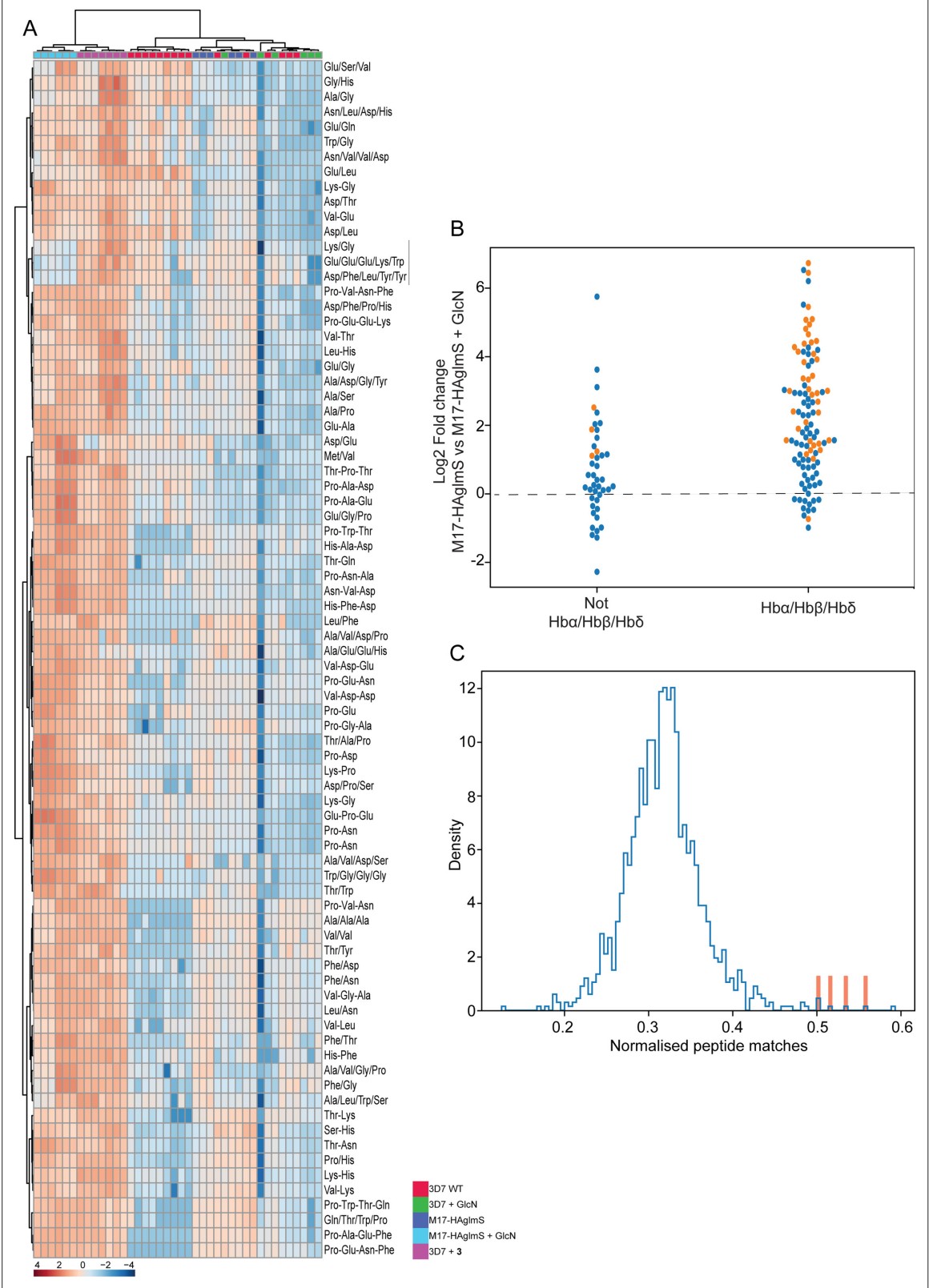

**Figure 9.** Targeted analysis of common significantly perturbed peptides (*P*-value <0.05) following addition of GlcN or treatment with **3** identified from experiment 1, 2 and 3. (**A**) Hierarchical clustering of the 80 significantly perturbed common peptides (fold change >1.5 and p-value <0.05) identified across the three independent experiments. Vertical clustering displays similarities between samples, while horizontal clusters reveal the relative abundances (median normalized) of the 80 peptides. The color scale bar represents log$_2$ (mean-centered and divided by the standard deviation of each

*Figure 9 continued on next page*

*Figure 9 continued*

variable) intensity values. Black bar indicates peptides increased in **3** treatment only. Peptides with hyphen (-) notations indicate confirmed sequence by MS/MS. Peptides with slash (/) notation indicate putative amino acid composition (accurate mass), without confirmed sequence order. (**B**) Differential enrichment of the 80 common significantly perturbed peptides that could or could not (blue dots) be derived from hemoglobin α, β, and δ. Orange dots are peptides that have MS/MS spectrum and their sequence have been confirmed. (**C**) Histogram of the sequence similarity of ~4700 proteins present in *P. falciparum*-infected erythrocytes to the peptides significantly dysregulated by *Pf*A-M17 disruption or inhibition. Here, sequence similarity is quantified as the number of times a significantly perturbed peptide matches a given protein, normalized by protein length. The Hb chains α, α2, β, and δ are highlighted in red.

latter since some of the peptides that were upregulated upon *Pf*A-M17 knockdown did not map to hemoglobin, this appears not to be the overriding function of *Pf*A-M17 and instead the different stages at which parasites stall in the presence of the activity-based probe are likely to stem from off-target effects.

Teasing out the function of *Pf*A-M17 has been further complicated by the fact that another aminopeptidase, *Pf*A-M1, has also been implicated in the final stages of hemoglobin digestion. Initial studies localized *Pf*A-M1 to the digestive vacuole, leading to speculation that the two aminopeptidases play a similar role in different parasite compartments (*Dalal and Klemba, 2007*). However, it is now clear that *Pf*A-M1 is also cytoplasmic (*Mathew et al., 2021*). Interestingly, inhibition of *Pf*A-M1 using an affinity-based probe resulted in parasites stalling at the trophozoite stage (*Harbut et al., 2011*), not dissimilar to the phenotypes observed herein upon knockdown of *Pf*A-M17 or treatment with **3**. However, treatment with this *Pf*A-M1 affinity-based probe also resulted in swelling of the digestive vacuole, a phenotype that we did not observe with *Pf*A-M17 knockdown; this phenotype has previously been seen when enzymes within the digestive vacuole implicated at the earlier stages of hemoglobin digestion have been targeted by compounds (*Harbut et al., 2011*; *Rosenthal et al., 1988*). Given that *Pf*A-M17 and *Pf*A-M1 are both cytoplasmic proteins, this then begs the question whether there is redundancy between the two, as has previously been shown with other enzymes involved in hemoglobin digestion such as falcipains and plasmepsins (*Liu et al., 2006*). Our results, however, indicate that loss of *Pf*A-M17 cannot be compensated for and as *Pfa-m1* has also been previously shown to be impervious to gene knockout, this would suggest that both aminopeptidases are required for parasite survival (*Dalal and Klemba, 2007*; *Zhang et al., 2018*). A thorough analysis of the P1 substrate specificities of recombinant *Pf*A-M17 and *Pf*A-M1 gives some indication as to why this may be. *Pf*A-M1 is a monomeric clan MA alanyl aminopeptidase with broad substrate specificity (*Poreba et al., 2012*). In contrast, *Pf*A-M17 is a clan MF leucine aminopeptidase with a far narrower specificity for the hydrophobic amino acids with a strong preference for a P1 leucine and tryptophan residue (*Poreba et al., 2012*). The inhibitor **3** takes advantage of this specificity with a biphenyl ring system bound in the S1 pocket satisfying the hydrophobic preferences of *Pf*A-M17 but with an addition of a hydroxymethyl group to allow the inhibitor to reach beyond the S1 pocket and pick up an interaction with the S-atom of Met392 (*Figure 4—figure supplement 1*). Taken together, the inhibitor specificity combined with the P1 substrate preference for leucine suggests *Pf*A-M17 may be a dedicated aminopeptidase for processing leucine, with leucine being the most abundant amino acid in human hemoglobin (*Hill et al., 1962*). Investigation of peptides identified in our metabolomic experiments, however, does not reveal an abundance of leucine-containing peptides, a result that is unexpected. The substrate specificity analysis performed *in vitro* is specific but cannot account for pathway dynamics or how the enzyme functions in a biological system and this must be taken into consideration. Using a conditional knockdown system negates any potential off-target effects and so the identified peptide sequences are indeed specific to the loss of *Pf*A-M17. As there is also aberrant DV formation after depletion of *Pf*A-M17, it is possible that there is an unidentified feedback mechanism that affects hemoglobin digestion upstream of *Pf*A-M17 in a manner that does not increase the quantity of hemozoin produced, leading to the diverse peptides that were observed by metabolomics. Although hemozoin was observed in the multiple DVs, vacuole integrity/lack of maturation and function is also potentially impacted and could lead to perturbed hemoglobin digestion and peptide production. For example, peptides originating from hemoglobin are required to be transported out of the DV into the cytoplasm for further cleavage and this could be impacted by aberrant digestive vacuole formation. A buildup of hemoglobin-derived peptides in the cytoplasm that cannot be processed by *Pf*A-M17 in the cytoplasm may also lead to a dysregulation of other aminopeptidases.

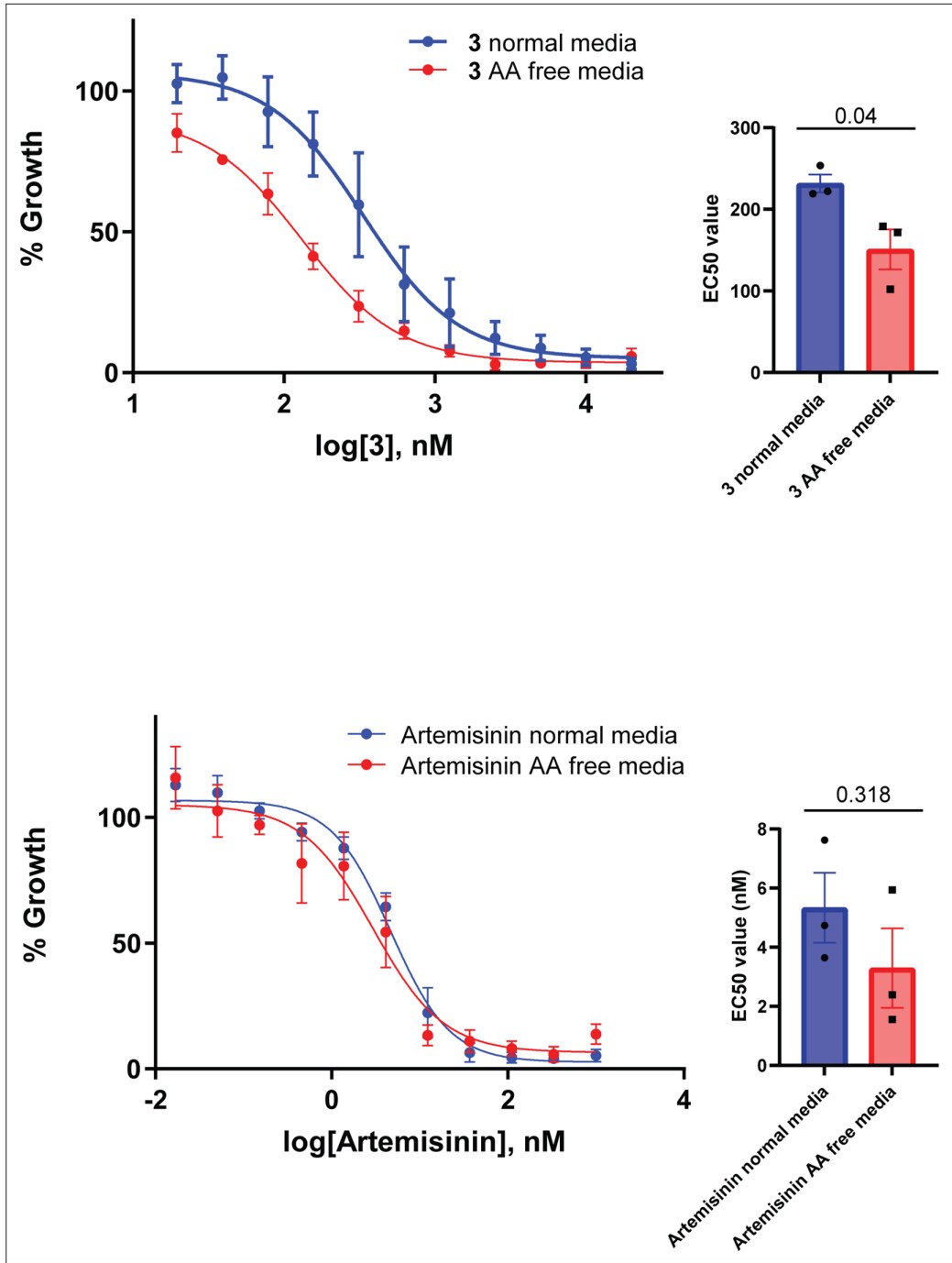

**Figure 10.** Removal of exogenous amino acids except for isoleucine sensitizes parasites to **3**. Killing action of **3** or artemisinin in either normal medium containing all amino acids (blue) or amino acid (AA) free medium, containing isoleucine (red) was measured over 72 hr and determined by SYBR Green I assay. The $EC_{50}$ values were calculated from three biological replicates performed in triplicate and data plotted as the mean ± standard error of the mean, with the inlay bar graphs showing the mean $EC_{50}$ values of these replicates with statistical significance determined using an unpaired t-test.

We have shown that selectivity of **3** for *Pf*A-M17 over *Pf*A-M1 is excellent against recombinant protein but cannot rule out whether there is any inhibition of *Pf*A-M1 within the parasite, particularly as there is a significant increase in the $EC_{50}$ when compared to the $K_i$ value (>300 nM compared to 18 nM). The use of **3** at high concentrations may also compound this effect, as 10 x treatments are approaching the $K_i$ values determined against *Pf*A-M1. However, unbiased thermal proteomics

profiling using concentrations upwards of 3 x the concentration expected to inhibit recombinant *Pf*A-M1 did not stabilize this protein, and over the different concentrations used only *Pf*A-M17 was identified to be significantly enriched and thus the target of **3**. Differences in the level of potency of an inhibitor against recombinant enzyme and parasite growth is a phenomenon that has been previously described and could be due to potential solubility problems or the ability of the compound to cross many membranes before it can interact with its target (*Mills et al., 2021*). This further highlights the importance of using compounds in conjunction with genetic editing techniques to confirm that phenotypes seen are not due to off-target effects. Another important consideration for compound development against *Pf*A-M17 will be the possible off-target effects against human aminopeptidases. Currently, the human leucyl aminopeptidases are poorly characterised and the further development of *Pf*A-M17 inhibitors as potential antimalarial therapeutics will need to take these into consideration.

Since our results indicate that *Pf*A-M17 digests peptides derived from hemoglobin, it is not surprising that this enzyme is not essential in the closely related apicomplexan parasite, *Toxoplasma gondii*, which does not infect red bloods cells and therefore is not exposed to hemoglobin (*Zheng et al., 2015*). However, the homologous leucine aminopeptidase is also not essential in the rodent malaria species, *P. berghei* (*Lin et al., 2015*). Analysis of the *Pb*A-M17 recombinant enzyme identified that it had a similar substrate specificity to that of *Pf*A-M17; however, there were notable differences in substrate specificity of the *Pb*A-M1 vs *Pf*A-M1 enzymes, suggesting a difference in aminopeptidase substrates between the two species (*Malcolm et al., 2021*). Knockout of *Pba-m17* did, however, result in a delay in parasite growth, although the significance of this cannot be inferred given the small sample number of two mice (*Lin et al., 2015*). Notably, isoleucine is present in mouse hemoglobin but absent from human hemoglobin. In *P. falciparum* - infected red blood cells, the influx of extracellular isoleucine can be mediated by leucine, which serves as a substrate of the isoleucine transporter at the parasite membrane (*Martin and Kirk, 2007*; *Poreba et al., 2012*). Since *Pf*A-M17 has been demonstrated to have considerable specificity for leucine (*Poreba et al., 2012*) and *P. berghei* can source isoleucine from hemoglobin, this may explain why *Pf*A-M17 is essential for *P. falciparum* survival whilst *P. berghei* can survive without *Pb*A-M17.

A novel finding observed after specific depletion of *Pf*A-M17 was the presence of multiple membrane-bound digestive vacuoles per parasite, a phenotype not previously seen with inhibition of aminopeptidases in *P. falciparum*. A limitation of transmission electron microscopy used here is that sections are only 2-D, but it is clear that the DV formation is severely affected after the loss of *Pf*A-M17. It is unclear whether this is because parasites attempt to endocytose more red blood cell cytoplasm to salvage their free amino acids pool and further investigation is required in order to elucidate the mechanism of this finding. As hemoglobin digestion begins during the ring stage, it is possible that this phenotype also stems from smaller endocytic vesicles failing to fuse to create the large digestive vacuole usually visible at trophozoite stage (*Abu Bakar et al., 2010*). We did not see a significant increase in the amount of hemozoin produced between parasites expressing or depleted of *Pf*A-M17, and a similar finding was observed between *P. berghei* wildtype and *pba-m17* knockout parasites (*Lin et al., 2015*). This indicates that depletion or loss of leucine aminopeptidases does not result in a significant overall increase in hemoglobin digestion resulting in altered hemozoin production.

In conclusion, we have demonstrated that *Pf*A-M17 plays an essential role in the survival of *P. falciparum* and is likely playing a major role in the release of amino acids originating from hemoglobin, confirming *Pf*A-M17 as a promising target for future antimalarial drugs. Further analysis into additional metabolites found not to map to hemoglobin will possibly provide insight into their origin and clues as to additional function(s) of *Pf*A-M17. Encouragingly, we have found that **3** is able to kill parasites in a sub-micromolar range, and we were able to confirm *Pf*A-M17 as the compound's target through unbiased thermal stability challenges. As this compound is most effective against the trophozoite stage, *Pf*A-M17 is a potential attractive partner target for ring stage antimalarial drugs. Now validated, **3** also provides scope for the further analysis into the function of *Pf*A-M17, negating problems that arise with the death phenotype of the knockdown and providing a rationale for further development of inhibitors against *Pf*A-M17.

# Materials and methods

## Chemistry

*Methyl 2-(4'-(hydroxymethyl)-[1,1'-biphenyl]–4-yl)–2-pivalamidoacetate (**2**).* To a mixture of methyl 2-(4-bromophenyl)–2-pivalamidoacetate (400 mg, 1.2 mmol) and 4-(hydroxymethyl)phenylboronic acid (222 mg, 1.5 mmol) in THF (6 mL) was added $Na_2CO_3$ (1 M, 2.0 eq). A steady stream of nitrogen was bubbled through the mixture for 5 min, before $PdCl_2(PPh_3)_2$ (0.03 eq) was added. The mixture was heated at 100 °C in a sealed tube for 2 hr. After cooling, the mixture was diluted with EtOAc (10 mL) and water (10 mL), and the aqueous layer discarded. The organic layer was concentrated under reduced pressure and then purified by flash chromatography to give the title compound (385 mg, 89%). $^1H$ NMR (DMSO-$d_6$) δ 7.54 (d, *J*=8.3 Hz, 2 H), 7.50 (d, *J*=8.2 Hz, 2 H), 7.40 (d, *J*=8.3 Hz, 2 H), 7.38 (d, *J*=8.3 Hz, 2 H), 6.87 (d, *J*=6.8 Hz, 1 H), 5.57 (d, *J*=6.8 Hz, 1 H), 4.65 (s, 2 H), 3.70 (s, 3 H), 1.24 (s, 9 H); $^{13}C$ NMR (DMSO-$d_6$) δ 178.1, 171.5, 141.0, 140.6, 139.1, 135.3, 127.5, 127.4, 127.3, 126.9, 64.2, 56.1, 52.7, 38.5, 27.2; LC-MS $t_R$: 3.2 min, *m/z* 356.0 [MH]$^+$.

*N-(2-(Hydroxyamino)–1-(4'-(hydroxymethyl)-[1,1'-biphenyl]–4-yl)–2-oxoethyl)pivalamide (**3**).* Methyl 2-(4'-(hydroxymethyl)-[1,1'-biphenyl]–4-yl)–2-pivalamidoacetate (**2**) (180 mg, 0.51 mmol) and $NH_2OH$. HCl (8.0 eq) were dissolved in anhydrous MeOH. KOH (5 M in MeOH, 10 eq) was added and the reaction mixture was stirred at RT overnight. After evaporation of the solvent, the crude product was purified by flash chromatography (eluent MeOH/DCM 0:100 to 10:90) to give the title compound as a white solid (155 mg, 86%). $^1H$ NMR (DMSO-$d_6$) δ 11.00 (s, 1 H), 9.05 (s, 1 H), 7.70 (d, *J*=8.0 Hz, 1 H), 7.63 (d, *J*=6.3 Hz, 2 H), 7.61 (d, *J*=6.2 Hz, 2 H), 7.48 (d, *J*=8.3 Hz, 2 H), 7.40 (d, *J*=8.2 Hz, 2 H), 5.40 (d, *J*=8.0 Hz, 1 H), 5.20 (t, *J*=5.7 Hz, 1 H), 4.53 (d, *J*=5.7 Hz, 2 H), 1.16 (s, 9 H); $^{13}C$ NMR (DMSO-$d_6$) δ 177.0, 166.9, 141.9, 139.6, 138.3, 138.1, 127.5, 127.2, 126.6, 126.5, 62.8, 53.7, 38.3, 27.3; *m/z* HRMS (TOF ES$^+$) $C_{20}H_{25}N_2O_4$ [MH]$^+$ calcd 357.1809; found 357.1818; LC-MS $t_R$: 3.0 min; HPLC $t_R$: 5.0 min, >99%. The cLogP was calculated using the ChemAxon chemistry cartridge via JChem for Excel software (version 16.4.11).

## Aminopeptidase activity assays using recombinant purified protein

Recombinant *Pf*A-M17 (amino acids 84–605) and *Pf*A-M1 (amino acids 85–605) expressed in *Escherichia coli* and purified by metal affinity chromatography followed by size-exclusion gel filtration was described previously (*Vinh et al., 2019*). Aminopeptidase activity was assessed by fluorescence assays using the fluorogenic peptide L-leucine-7-amido-4 methylcoumarin hydrochloride (Sigma L2145) as a substrate as previously described (*Vinh et al., 2019*). Michaelis−Menten constants ($K_m$) were calculated for each enzyme purification and showed similar activity as reported previously (*McGowan et al., 2010*; *McGowan et al., 2009*). Inhibition of aminopeptidase activity was measured using a Morrison inhibition constant ($K_i^{(app)}$), where enzymes were preincubated in 100 mM Tris−HCl, pH 8.0 (supplemented with 2 mM $CoCl_2$ for *Pf*A-M17) and compound **3** for 10 min prior to the addition of substrate (20 µM for *Pf*A-M1, 10 µM for *Pf*A-M17). Substrate concentrations did not exceed the $K_m$ for each enzyme. The inhibitor concentration range was selected to obtain a complete inhibition curve (0–100%). $K_i^{(app)}$ values were calculated by plotting the initial rates versus inhibitor concentration, and fitting to the Morrison equation for tight-binding inhibitors in GraphPad Prism software (nonlinear regression method). Inhibition constants were calculated in biological triplicate from three different protein preparations. $K_i^{(app)}$ value represents the mean and standard error of the mean (SEM).

## Structural biology

*Pf*A-M17 was co-crystallized with **3** by the hanging-drop method, using previously established protocols (*Vinh et al., 2019*). Briefly, *Pf*A-M17 was concentrated to 10 mg/mL and co-crystallized with a final ligand concentration of 1 mM in 30–40% PEG400, 0.1 M Tris pH 7.5–8.5, 0.2 M $Li_2SO_4$. Crystals were soaked overnight in mother liquor supplemented with 1 mM ligand and 1 mM $ZnSO_4$ before being harvested for data collection. Crystals were snap-frozen in liquid nitrogen, and data were collected 100 K using synchrotron radiation at the Australian Synchrotron beamlines 3ID1(MX2). Data was processed using XDS and Aimless as part of the CCP4i program suite. The structures were solved by molecular replacement in Phaser using RCSB ID 3KQZ as the search model. The structures were refined using Phenix with 5% of reflections set aside for calculation of R$_{free}$. Between refinement cycles, the protein structure, solvent, and inhibitor were manually built into *2Fo − Fc* and *Fo − Fc* electron density maps using COOT with restraint files generated by Phenix where necessary. Data collection

and refinement statistics can be found in *Supplementary file 1*. The coordinates and structure factors are available from the Protein Data Bank with PDB accession code 7RIE (*Plasmodium falciparum* M17 in complex with inhibitor MIPS2571, Authors: Chaille T Webb and Sheena McGowan).

## Antibody production

Recombinant *Pf*A-M17 (amino acids 84–605) (*Vinh et al., 2019*) was used to generate polyclonal rabbit antibodies using The Walter and Eliza Hall Institute of Medical Research Antibody Facility. Briefly, a prebleed was collected from rabbits prior to subcutaneous immunization on four occasions with 200 µg of protein, the first in Freund's complete adjuvant and the subsequent immunizations in incomplete adjuvant. The final rabbit bleed (and prebleed for comparison) was used to probe for *Pf*A-M17.

## Plasmid constructs

A transgenic *P. falciparum* line allowing conditional knockdown of *Pf*A-M17 was generated by inserting the last 1017 base pairs of *Pf*A-M17 (Pf3D7_1446200), excluding the stop codon, into the *Bgl*II and *Pst*I sites of pPfTEX88-HAglmS using oligonucleotides DO733 and DO734 (*Chisholm et al., 2016*). This resulted in the fusion of the *Pf*A-M17 C-terminus with a triple hemagglutinin (HA) and single streptavidin epitope tag. Oligonucleotide sequences are provided in *Supplementary file 2*.

## Parasite culture and transfection

*P. falciparum* 3D7 and transgenic lines were cultured continuously (*Trager and Jensen, 1976*) in O$^+$ human erythrocytes obtained from the Australian Red Cross at 4% hematocrit. Complete culturing media contained RPMI 1640 medium (Life Technologies), 20 mg/L gentamicin, 50 mg/L hypoxanthine, 25 mM sodium bicarbonate, 25 mM HEPES and 0.5% (w/v) Albumax II (Life Technologies). Minimal media (ResolvingImages) contained the same constituents as complete media except that isoleucine was the only amino acid present. Parasite cultures were maintained at 37 °C at atmospheric conditions of 5% CO$_2$ and 1% O$_2$ in N$_2$. Transfections were performed as previously described (*Fidock and Wellems, 1997*) and selected for with 2.5 nM WR99210. Parasite cultures were visualized by making thin blood smears on glass slides, which were fixed with methanol for 10 s and then stained with Giemsa (1:10 dilution in water; Merck) for 5–10 min prior to microscopy under ×100 magnification with oil.

## Analysis of *Pf*A-M17 expression over the erythrocytic lifecycle and western blotting

Erythrocytes infected with wildtype *Pf*3D7 parasites were heparin synchronized as described (*Boyle et al., 2010*) and allowed to reinvade for 4 hr before sorbitol synchronization (*Lambros and Vanderberg, 1979*) to remove remaining schizonts. Parasites were cultured for a cycle and samples were taken at six hourly intervals and lysed in 0.1% (w/v) saponin. Parasite lysates were separated on 4–15% Mini-PROTEAN TGX Gels (Biorad) and transferred to a nitrocellulose membrane for Western blotting. After blocking in 3% (w/v) bovine serum albumin (BSA) in PBS, the membrane was incubated with rabbit anti-M17 (1:1000) and rabbit anti-HSP101 (1:1000; *de Koning-Ward et al., 2009*) as a loading control. After washing, the blots were incubated with horseradish peroxidase-conjugated secondary antibodies (1:10,000; Thermo Scientific). Protein bands were detected using the Clarity ECl Western blotting substrate (Biorad) and imaged using a Fujifilm LAS-4000 Luminescent Image Analyzer. ImageJ software (NIH, version 151 r) was used to measure the intensity of the bands and GraphPad Prism (V.8.4.2) was used to plot the densitometry. Western blotting to confirm expression of epitope tagged *Pf*A-M17 was performed on asynchronous parasite lysates using rabbit anti-M17 (1:1000) and rabbit anti HSP70 (1:1000).

## Sequential solubility assay

Erythrocytes infected with *Pf*3D7 parasites at trophozoite stage were lysed in 0.05% (w/v) saponin in PBS containing a complete protease inhibitor cocktail (Sigma-Aldrich). Sequential solubilization was performed as previously described (*Counihan et al., 2017*). Briefly, the parasite pellet was resuspended in a hypotonic lysis buffer (1 mM HEPES, pH 7.4) and incubated on ice for 30 min before undergoing three rounds of freeze-thawing in liquid nitrogen. The solution was then centrifuged at

100,000 $g$ for 30 min at 2 °C and the supernatant, containing the soluble proteins, collected for analysis. The pellet was then resuspended in 0.1 M $Na_2CO_3$ (pH 11.5), incubated for 30 min and centrifuged, followed by collection of the supernatant fraction, which contained membrane-associated proteins. The remaining pellet was resuspended in 1% (v/v) Triton X-100 in PBS and incubated at room temperature before undergoing centrifugation, with the supernatant containing integral membrane proteins. The final pellet, which contained the insoluble proteins, was resuspended in 1% (v/v) Triton X-100 in PBS. All samples underwent SDS-PAGE gel electrophoresis and western blotting, with membranes probed with the following antibodies: rabbit anti-M17 (1:1000), rabbit anti-GAPDH (1:1000), rabbit anti-HSP101 (1:1000 *de Koning-Ward et al., 2009*), and rabbit anti-EXP2 (1:1000 *Bullen et al., 2012*).

## Immunofluorescence analysis

Transgenic parasites were smeared onto glass slides and allowed to dry overnight before being fixed using acetone:methanol (90:10) for 2 min at –20 °C. Slides were air dried before being placed at –20 °C until analysis. For immunofluorescent assays, slides were thawed at 37 °C for 10 min and blocked with 1% (w/v) BSA for 1 hr. Primary antibody rat anti-HA (Life Technologies) was diluted 1:250 in 0.5% BSA and applied to slides for 2 hr before three 5 min washes in PBS. The appropriate AlexaFluor 488-conjugated secondary antibody (1:1000; Life Technologies) was diluted in 0.5% BSA and incubated on slides for 1 hr before being washed for 5 min three times in PBS. Cover slips were mounted using Prolong Gold Antifade reagent containing 4′,6-diamidino-2-phenylindole (DAPI; Life Technologies) and incubated overnight at 37 °C. Images were taken on a Nikon Eclipse Ti2 microscope at ×100 magnification under oil immersion and processed using ImageJ software (NIH, version 1.53 c).

## Knockdown of *Pf*A-M17 expression in *P. falciparum* and growth analysis

Heparin synchronized *Pf*M17-HAglmS parasites were treated with 2.5 mM glucosamine (GlcN) at 0–4 hr post-invasion; untreated parasites and *Pf*3D7 parasites treated with 2.5 mM glucosamine served as controls. Parasites were harvested in the first cycle of GlcN treatment (cycle 1) at trophozoite stage, as well as the cycle following (cycle 2) at the same stage. Harvested parasites were lysed in 0.05% saponin, separated by SDS-PAGE electrophoresis and knockdown was analyzed by western blotting using mouse anti-HA (1:1000; Sigma) antibody with rabbit anti-EXP2 (1:1000) as a loading control. Assessment of parasite growth ±GlcN was determined by analysis of Giemsa-stained smears and images taken with a SC50 5-megapixel or IX71 color camera (Olympus). The parasitemia was determined daily for 6 days following ± GlcN treatment by counting a minimum of 1000 cells. For each day, the stage of development was also determined and plotted on GraphPad Prism (V.8.4.1). Survival of parasites seeded in 96-well plates at 100 parasites per well was determined after 10 days in culture by Sybr Green I assay. Briefly, after freeze-thawing at –80 °C, an equal volumes of Lysis buffer (20 mM Tris pH 7.5, 5 mM EDTA, 0.008% saponin (w/v) & 0.008% Triton x-100 (v/v)) (*Smilkstein et al., 2004*) containing 0.2 µL/mL SYBR Green I Nucleic Acid Gel Stain (10,000 x in DMSO; ThermoFisher) was added to each well and incubated for 1 hr at RT before fluorescence intensity was read on a Glomax Explorer Fully Loaded (Promega) with emission wavelengths of 500–550 nm and an excitation wavelength of 475 nm and graphs were generated using GraphPad Prism (V.8.4.1). All experiments were performed in 3 biological replicates and unless otherwise stated, a two-tailed unpaired t-test was used herein to determine statistical significance, with data presented as the mean and error bars representative of standard deviation.

## Hemozoin assay

Erythrocytes infected with *Pf*M17-HAglmS or *Pf*3D7 underwent sorbitol synchronization before cultures were treated ±GlcN and incubated for a further cycle until parasites developed into trophozoites. Once parasites reached developmentally similar stages, pellets were resuspended in 800 µL of 2.5% SDS in 0.1 M sodium bicarbonate pH 8.8, and then incubated at RT with rotation for 20 min, before undergoing centrifugation at 13,000 $g$ for 10 min. Each pellet was then washed twice with 1 mL of the same solution before being resuspending in 500 µL of 5% SDS, 50 mM NaOH and incubating for a further 20 min with rotation. The quantity of monomeric heme was then measured at 405 nM on a Glomax Explorer Fully Loaded (Promega). Three or four biological replicates were performed, and the data plotted using GraphPad Prism 9 where significance was determined by a two-tailed unpaired

t-test. A one-way ANOVA and Dunn's Multiple Comparison test was used to determine the significance of the different numbers of digestive vacuoles per parasites.

## Determination of compound EC$_{50}$

Parasite viability assays were adapted from previously described methods (*Dery et al., 2015*). Briefly, sorbitol synchronized ring stage *Pf*3D7 parasites were cultured in 96-well U-bottom plates at 0.5% parasitemia and 2% hematocrit, to which 50 µL of serially diluted **3,** artemisinin or artesunate (Sigma) was added. After 72 hr under standard culturing conditions, plates were placed at –80 °C before analysis using the SYBR Green I assay as described above. Uninfected RBCs and parasites treated with the vehicle control DMSO were used to normalize fluorescence. Data from three or four biological replicates performed in triplicate was plotted as four-parameter log dose nonlinear regression analysis with a sigmoidal dose-response curve fitted using GraphPad Prism 9 to generate the EC$_{50}$ values, with error bars representative of the SEM.

## Parasite killing rate assay

Assay was performed as previously described with some alterations (*Gilson et al., 2019*). Briefly, sorbitol synchronized ring or trophozoite stage parasites were cultured in the presence of 10 x the EC$_{50}$ of **3** or artesunate for either 24 or 48 hr. Cultures that were incubated with either compound for 48 hr were fed at 24 hr with fresh media containing 10 x the EC$_{50}$ of compound. At the completion of these times, RBCs were thoroughly washed to remove the compound and cultures were diluted 1/3 with fresh media and further grown for 48 hr before aliquots were placed at –80 °C. Cultures were thawed and analyzed using SYBR Green I assay as described above. Parasite viability was determined as a percentage of DMSO-treated parasites cultured alongside compound treated parasites. Artesunate was used as a positive control and for comparison of the parasite killing rate, and experiments were performed in four biological replicates.

## Transmission electron microscopy

*Pf*3D7 parasites treated with 10 x EC$_{50}$ of **3** or the same concentration of vehicle control (DMSO) were spun in Eppendorf tubes in a Thermo Scientific Heraeus Megafuge 40 centrifuge at 2000 rpm for 3 min to pellet the RBC. The resulting pellets were fixed in 2.5% glutaraldehyde in 0.1 M Sorensen's phosphate buffer (pH 7.2, 300 mOsmol) at 4 °C. Taking care not to completely resuspend the RBC pellets, fragments of the pellets were intentionally produced by gently surging with analogous buffer delivered via pipette tip. Due to the fragility of the fragments, gentle inversion of the Eppendorf tubes was used to mix reagents during processing rather than agitation. After washing the fragments in buffer for 15 min, they were immersed in a secondary fixative of 1% osmium tetroxide in buffer for 60 min. The fragments were dehydrated in a graded ethanol series, infiltrated with absolute dry ethanol mixed with Spurr's Resin for 30 min and then two changes of 100% Spurr's Resin for 60 min. Singular fragments were embedded in resin in BEEM polyethylene capsules and polymerized overnight at 65 °C. Ultrathin sections were stained with uranyl acetate, followed by lead citrate. Sections were examined using a JEOL TEM – 1400 120kV TEM and images acquired with a Gatan UltraScan 1000 camera.

## Sample preparation for metabolomics analysis

For experiment 1 and 2, heparin synchronized *Pf*3D7 and *Pf*M17-HAglmS parasites were allowed to invade RBC for 4 hr and any remaining schizonts were lysed by sorbitol synchronization. For experiment 1, parasite cultures were then treated for ~36 hr with 2.5 mM GlcN or for 1 hr with **3** at 10 x the EC$_{50}$ (*Pf*3D7 only) or left untreated, while for experiment 2, parasite cultures were only treated with GlcN. Parasites were harvested at developmentally similar timepoints by centrifugation at 900 *g* for 5 min and then resuspended in 10 mL of chilled PBS. Parasite metabolism was quenched by cooling samples to between 3°C and 5°C in an ethanol-dry ice bath. The rest of the preparation was performed at 4 °C. Parasites were magnet purified on a VarioMACS column and 3x10$^7$ parasites were used for downstream analysis. For experiment 3, *Pf*3D7 cultures underwent double sorbitol synchronization 14 hr apart, followed by further incubation for 28–42 hr to achieve the desired trophozoite stage (28 hpi) at 6% parasitaemia and 2% hematocrit. Infected RBCs (2x10$^8$) were treated with 10 x the EC$_{50}$ of **3** for 1 hr, after which metabolites were extracted. All samples (from experiment 1, 2 and

3) were centrifuged at 650 *g* for 3 min, the supernatant was removed, and the pellet washed in 500 μL of ice-cold PBS. Samples were again centrifuged at 650 *g* for 3 min and pellets were resuspended in 150 μL of ice-cold extraction buffer (100% methanol) and quickly resuspended. The samples were then incubated on a vortex mixer for 1 hr at 4 °C before being centrifuged at 17,000 *g* for 10 min; from this 100 μL of supernatant was collected and stored at –80 °C until analysis. For each sample, another 10 μL was collected and pooled, to serve as a quality control (QC) sample.

## Liquid chromatography- mass spectrometry (LC-MS) analysis

Liquid chromatography-mass spectrometry (LC-MS) data was acquired on a Q-Exactive Orbitrap mass spectrometer (Thermo Scientific) coupled with high-performance liquid chromatography system (HPLC, Dionex Ultimate 3000 RS, Thermo Scientific) as described previously (*Creek et al., 2016*). Briefly, chromatographic separation was performed on ZIC-pHILIC column equipped with a guard (5 μm, 4.6×150 mm, SeQuant, Merck). The mobile phase (A) was 20 mM ammonium carbonate (Sigma Aldrich), (B) acetonitrile (Burdick and Jackson) and needle wash solution was 50% isopropanol. The column flow rate was maintained at 0.3 mL/min with temperature at 25 °C and the gradient program was: 80% B to 50% B over 15 min, then to 5% B at 18 min until 21 min, increasing to 80% B at 24 min until 32 min. Total run time was 32 min with an injection volume of 10 μL. Mass spectrometer was operated in full scan mode with positive and negative polarity switching at 35,000 resolution at 200 *m/z* with detection range of 85–1275 *m/z*, AGC target was 1e6 ions with a maximum injection time of 50ms. Electro-spray ionization source (ESI) was set to 4.0 kV voltage for positive and negative mode, sheath gas was set to 50, aux gas to 20 and sweep gas to 2 arbitrary units, capillary temperature 300 °C, probe heater temperature 120 °C. Approximately 350 authentic metabolite standards were analyzed at the start of each batch to provide accurate retention times to facilitate metabolite identification. Metabolomics samples were analyzed as a single batch in random order with periodic injections of the pooled QC, and blank samples, to assess analytical quality and aid downstream metabolite identification procedures.

## Metabolomics LC-MS data processing

The acquired LCMS data was processed in untargeted fashion using open source software, IDEOM (*Creek et al., 2012*) (http://mzmatch.sourceforge.net/ideom.php) using an updated metabolite database containing all peptides of proteinergic amino acids up to five amino acids in length. Initially, *ProteoWizard* was used to convert raw LC-MS files to *mzXML* format and *XCMS* (Centwave) to pick peaks and convert to *peakML* files. *Mzmatch.R* was used for alignment and annotation of related metabolite peaks with a minimum detectable intensity of 100,000, relative standard deviation (RSD) of <0.5 (reproducibility), and peak shape (codadw) of >0.8. Default IDEOM parameters were used to eliminate unwanted noise and artefact peaks. Loss or gain of a proton was corrected in negative and positive ESI mode, respectively, followed by putative identification of metabolites by accurate mass within 3 ppm mass error searching against the IDEOM metabolite database. To reduce the number of false positive identifications, retention time error was calculated for each putative ID using IDEOM's built-in retention time model which uses actual retention time data of authentic standards (~350 standards). Furthermore, the identification of a set of peptides was based on MS/MS analysis, which allowed definitive confirmation of the amino acid sequence in selected peptides. Principal-component analysis (PCA) and hierarchical clustering algorithms were run also in Metaboanalyst (*Chong et al., 2018*). Metabolomics data are presented as relative abundance values from three technical replicates for experiment 1 and 2, while for experiment 3, data is from four to nine biological replicates. Differences were determined using Welch's t test where significant interactions were observed. Significance was determined at p-values <0.05. To assess whether identified peptides could be derived from hemoglobin, the sequences of human hemoglobin α, β and δ chains (HBA_HUMAN P69905, HBB_HUMAN P68871, HBD_HUMAN P02042) were searched for any peptide matching the MS/MS-derived sequence, or any peptide with monoisotopic mass within 0.002 *m/z* of the identified peptide using custom Python scripts (*MacRaild, 2022*) (https://github.com/macraild/Hb_peptide_analysis, copt archived at swh:1:rev:ce52dc4205dc1e5ec646611d1cbb103dd667f868). In the same way, we quantified the number of peptide matches for each of the 3113 *P. falciparum* and 1617 *H. sapiens* proteins that we recently identified in *P. falciparum*-infected erythrocytes (*Siddiqui et al., 2022*).

## Thermal proteomics profiling (TPP)

Parasites were isolated from iRBCs via hypotonic lysis (150 mM Ammonium chloride, 10 mM Potassium bicarbonate, 1 mM Ethylenediaminetetraacetic acid (EDTA) in Milli-Q water) for 5 min at 4°C. Following lysis, the pellets were then resuspended in 100 mM HEPES (4-(2-hydroxyethyl)–1-piperaz ineethanesulfonic acid) pH 8.1, and a freeze-thaw lysis was performed by cycling (3 x) between dry ice and a heating block set at 37°C for 3 min. Cell debris was pelleted by centrifuging at 20 000 $g$ for 20 min at 4°C and supernatant collected. Parasite lysate was separated into equal technical replicates (4) and these replicates were separated into different conditions, DMSO control and compound **3** at 3 µM and 12 µM. The samples were then incubated at room temperature for 3 min before being thermally challenged by heating at 60°C for 5 min. The denatured protein fraction was then removed via ultracentrifugation at 100,000 $g$ for 20 min at 4°C (Beckman Coulter Optima XE-90 – IVD ultracentrifuge with a 42.2 Ti rotor). The soluble fraction was reduced and alkylated using tris(2-carboxyethyl) phosphine hydrochloride (TCEP) (10 mM final concentration) and iodoacetamide (40 mM final concentration) (Sigma) and heated at 45°C for 15 min. Following alkylation and reduction, protein concentration was determined using a BCA assay, an appropriate concentration of trypsin (1:50; Promega) was added before incubation overnight at 37 °C. The following day, trypsin activity was quenched using 5% formic acid (FA) and samples were subjected to desalting using in-house-generated StageTips as described previously (*Rappsilber et al., 2003*). The samples were then dried and resuspended in 12 µL of 2% (v/v) acetonitrile (ACN) and 0.1% (v/v) FA containing indexed retention time (iRT) peptides (Biognosys) for LC-MS/MS analysis.

## LC-MS/MS and data analysis

LC-MS/MS was carried out using data-independent acquisition mode as described previously (*Siddiqui et al., 2022*). Raw files were then processed using SpectronautTM 13.0 against an in-house generated *P. falciparum* (*Pf*3D7 line) spectral library as described previously (*Siddiqui et al., 2022*). Following protein identification using the software SpectronautTM 13.0 as previously described (*Siddiqui et al., 2022*), proteins identified were exported as an excel sheet and their fold change of drug-treated conditions relative to the 0 µM control for each experiment was calculated (only for proteins with intensities greater than $1\times10^5$ and with a minimum peptide count of 2), and the significance of the change was determined by a Welch's t-test, with a p-value <0.05 deemed significant. Proteins that were significantly stabilized (fold change >1.15, p-value <0.05) at multiple concentrations of compound **3** were considered 'hits' and these were plotted using paired volcano plots.

## Acknowledgements

We thank the Australian Red Cross for providing red blood cells used in this study. We would also like to thank Professor Susan Charman and the Centre for Drug Candidate Optimisation (CDCO) team for providing compound analysis as well as Sandra Crameri for assistance with TEM. RCSE and TRM were recipients of an Australian Government Research Training Program Stipend. This work was supported by an NHMRC Synergy Grant (1185354). TFdK-W is the recipient of an NHMRC Fellowship (1136300).

## Additional information

### Funding

| Funder | Grant reference number | Author |
| --- | --- | --- |
| National Health and Medical Research Council | 1185354 | Darren J Creek Peter J Scammells Sheena McGowan Tania F de Koning-Ward |
| National Health and Medical Research Council | 1136300 | Tania F de Koning-Ward |
| Australian Government | Research Training Program Stipend | Rebecca CS Edgar Tess R Malcolm |

| Funder | Grant reference number | Author |
| --- | --- | --- |

The funders had no role in study design, data collection and interpretation, or the decision to submit the work for publication.

## Author contributions

Rebecca CS Edgar, Conceptualization, Formal analysis, Investigation, Writing – original draft, Writing – review and editing; Ghizal Siddiqui, Formal analysis, Investigation, Writing – original draft, Writing – review and editing; Katheryn Hjerrild, Tess R Malcolm, Investigation; Natalie B Vinh, Conceptualization, Investigation; Chaille T Webb, Formal analysis; Clare Holmes, Willy W Suen, Formal analysis, Investigation, Writing – original draft; Christopher A MacRaild, Formal analysis, Writing – original draft, Writing – review and editing; Hope C Chernih, Formal analysis, Investigation; Natalie A Counihan, Formal analysis, Supervision, Investigation, Writing – review and editing; Darren J Creek, Supervision, Writing – original draft, Writing – review and editing; Peter J Scammells, Conceptualization, Supervision, Writing – original draft; Sheena McGowan, Conceptualization, Formal analysis, Supervision, Investigation, Writing – original draft, Writing – review and editing; Tania F de Koning-Ward, Conceptualization, Formal analysis, Supervision, Funding acquisition, Investigation, Writing – original draft, Project administration, Writing – review and editing

## Author ORCIDs

Rebecca CS Edgar ⓘ http://orcid.org/0000-0002-5974-8625
Ghizal Siddiqui ⓘ http://orcid.org/0000-0002-3153-3198
Hope C Chernih ⓘ http://orcid.org/0000-0003-2620-9518
Natalie A Counihan ⓘ http://orcid.org/0000-0002-8973-3344
Darren J Creek ⓘ http://orcid.org/0000-0001-7497-7082
Peter J Scammells ⓘ http://orcid.org/0000-0003-2930-895X
Sheena McGowan ⓘ http://orcid.org/0000-0001-6863-1106
Tania F de Koning-Ward ⓘ http://orcid.org/0000-0001-5810-8063

## Decision letter and Author response

Decision letter https://doi.org/10.7554/eLife.80813.sa1
Author response https://doi.org/10.7554/eLife.80813.sa2

# Additional files

## Supplementary files

• Supplementary file 1. Crystallography and refinement statistics for *Pf*A-M17 bound to 3.

• Supplementary file 2. Oligonucleotide sequences used in this study.

• Transparent reporting form

• Source data 1. Source data for *Figures 3–7, 9 and 10*.

• Source data 2. Source data of metabolite abundance for *Figures 8 and 9* and associated figure supplements.

## Data availability

Source data has been provided for Figures 3, 4, 5, 6, 7, 9, 10 and Figure 3—figure supplement 2. Metabolomics data has been provided for Figures 8, 9, and Figure 8—figure supplements 1–3. Structural data has been deposited with PDB (ID 7RIE) for Figure 5 and Figure 4—figure supplement 1. The proteomics data from Figure 6 has been uploaded to the ProteomeXchange Consortium via the PRIDE partner repository with the dataset identifier PXD032358. Raw metabolomic data is available at the NIH Common Fund's National Metabolomics Data Repository (NMDR) website, the Metabolomics Workbench, https://www.metabolomicsworkbench.org, where it has been assigned Project ID (ST002106 for Exp 1, ST002107 for Exp 2, and ST002108 for Exp 3).

The following datasets were generated:

| Author(s) | Year | Dataset title | Dataset URL | Database and Identifier |
|---|---|---|---|---|
| Siddiqui G | 2022 | Genetic and chemical validation of *Plasmodium falciparum* aminopeptidase PfA-M17 as a drug target in the hemoglobin digestion pathway | http://dx.doi.org/10.21228/M85416 | Metabolomics Workbench, 10.21228/M85416 |
| Webb CT, McGowan S | 2022 | *Plasmodium falciparum* M17 in complex with inhibitor MIPS2571 | https://www.rcsb.org/structure/7RIE | RCSB Protein Data Bank, 7RIE |
| Siddiqui G | 2022 | Genetic and chemical validation of *Plasmodium falciparum* aminopeptidase PfA-M17 as a drug target in the hemoglobin digestion pathway | https://www.ebi.ac.uk/pride/archive/projects/PXD032358 | PRIDE, PXD032358 |

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
