## [Editor Report]

This article offers further characterization of PfA-M17, a *P. falciparum* aminopeptidase that has been studied for some years and was previously shown to be an essential protein predicted to function in late steps of hemoglobin hydrolysis by erythrocytic parasites. The new report adds valuable demonstration of impacts of PfA-M17 knockdown, description of the synthesis and characterization of a novel PfA-M17 inhibitor with high nanomolar activity against cultured parasites, and metabolomic analysis of inhibited parasites adding evidence for hemoglobin to be the primary target of the protease. The work is important to our understanding of the roles of aminopeptidases in parasite biology and the claims are convincingly supported by the data.

---

## [Decision Letter]

**Decision letter after peer review:**

[Editors’ note: the authors submitted for reconsideration following the decision after peer review. What follows is the decision letter after the first round of review.]

Thank you for submitting the paper "Genetic and chemical validation of *Plasmodium falciparum* aminopeptidase *Pf*A-M17 as a drug target in the hemoglobin digestion pathway" for consideration by *eLife*. Your article has been reviewed by 3 peer reviewers, and the evaluation has been overseen by a Reviewing Editor and a Senior Editor. The following individuals involved in review of your submission have agreed to reveal their identity: Michael J Blackman (Reviewer #1).

Comments to the Authors:

A decision has been reached after consultation between the reviewers. Based on these discussions and the individual reviews below, the reviewers concur that the work offers a sizable contribution to our understanding of the roles of aminopeptidases in parasite biology using genetic and pharmacologic approaches. However, they also shared some concerns and identified weaknesses that are summarized below. Some of the major points (1 and 3) as well as the minor issues raised in individual reviews need to be addressed experimentally prior to consideration for acceptance at *eLife*.

Given the extensive revision required (which will take > 2 months of additional work), the editorial guidelines for the journal dictate that I must reject this paper. However, should you decide to carry out the additional experiments and revise your manuscript as indicated by the referees, then we would be happy to consider a resubmission and handle it as a revision

1. The morphological phenotype observed both in PfA-M17 knockdown and in wt parasites during inhibitor treatment is interesting but the micrograph showing this phenotype is sub-optimal and needs to be consolidated. The authors should be inspired by reviewer suggestions to use lysosomotropic dye, such as lysotracker or to document the phenotype by electron microscopy.

2. The demonstration that a selective inhibitor of PfA-M17 kills the parasite *in vitro* lacks novelty, as similar findings have been published previously. Also, the co-crystal structure which illustrates that the enzyme accommodates the inhibitor in an expansive S1 subsite, has been previously shown for structurally related inhibitors.

3. Importantly the data addressing the specificity of inhibitor 3 is lacking. The rationale for adding the inhibitor when the knockdown parasites are already on their way to dying to demonstrate specificity is not convincing. Suggested approaches to validating specificity include identifying resistance mutations in the target, overexpressing the target and observing an increase in EC50, observing increased sensitivity to chemical inhibition following genetic knockdown, or generating a chemical probe for direct labeling.

4. The comparative metabolomics analysis leads to confounding results.

Given the strict specificity PfA-M17 (strong preference for a P1 leucine and tryptophan residue), the accumulating peptides are not consistent with loss of the specific proteolytic specificity of PfA-M17. Moreover, the comparison of the impact of PfA-M17 knockdown or compound 3 treatment show modest overlap. It is plausible that a wide perturbation of catabolism would result from a loss of vacuolar integrity.

5. The authors should discuss the risk of inhibition of host proteases and the potential limitation of the slow action of the inhibitor against cultured parasites.

*Reviewer #1 (Recommendations for the authors):*

There is a need to discover new targets for new kinds of drugs to tackle malaria, which is a major health threat across the globe. In this work, the authors took 3 complementary approaches to investigate the function, essentiality and druggability of an enzyme of the malaria parasite *Plasmodium falciparum*, an aminopeptidase called PfA-M17. First they used a conditional gene knockdown strategy to show that parasites depleted of PfA-M17 cannot proliferate in culture. Next, they designed a small molecule compound designed to inhibit PfA-M17 and showed that this compound (called compound 3) both inhibited recombinant PfA-M17 with sub-μM potency and killed the parasite, producing a phenotype similar to that resulting from gene knockdown. In efforts to demonstrate on-target efficacy of compound 3, the authors additionally showed that parasites depleted of PfA-M17 by gene knockdown are killed by compound 3 with similar kinetics (and with a similar phenotype) to parasites expressing wild-type levels of the enzyme. Finally, by comparing the metabolomic profile of gene knockdown and compound 3-treated parasites, the authors showed that a number of peptide products likely derived from haemoglobin accumulated in both parasite populations. Collectively these results support the on-target activity of compound 3 and provide evidence that PfA-M17 acts to degrade haemoglobin-derived peptides in vivo. The authors conclude that PfA-M17 is a potential target for a new type of antimalarial therapy based on inhibitors of the enzyme.

Strengths: Overall, the work is well-presented and uses an appropriate combination of orthogonal technologies to interrogate the function of this intriguing enzyme. The study makes a useful contribution to knowledge of basic biology of the parasite, but importantly also provides insights valuable for drug discovery efforts, important in the light of growing evidence for emerging artemisinin resistance in Africa. The phenotypic response to enzyme knockdown (genetic or chemical) by the apparent generation of multiple digestive vacuoles is intriguing and potentially very informative, but needs to be better supported experimentally.

Weaknesses: Whilst the majority of the conclusions are very well supported by the experimental data, a major aspect of the work – the demonstration that a selective inhibitor of PfA-M17 kills the parasite *in vitro* – lacks novelty, as similar findings have been published previously both by some of these authors and by others (e.g. PMID: 27449897, PMID: 26406322, PMID: 26807544). The conclusion that genetic knockdown of PfA-M18 or inhibition of the enzyme with compound 3 results in multiple digestive vacuoles would also be much better supported by electron microscopic analysis rather than only light microscopy as in the current manuscript.

Specific issues to be addressed experimentally:

1. The knockdown data provided in Figure 3 are good, but it would be more convincing to see these data supplemented by immunofluorescence data (ideally showing a full field of parasites) probing with anti-HA antibodies or anti-PfA-M17 antibodies. These data would be complementary to the western blot results in that they show expression levels in individual parasites.

2. The observation that genetic knockdown of PfA-M17 or treatment with compound 3 results in parasites exhibiting multiple haemozoin granules is very interesting indeed and may indeed shed light on the parasite's response to these perturbations, as speculated by the authors in the Discussion. However, in my view these light microscopic observations need to be supported by electron microscopic analysis in order to provide a definitive evaluation of this phenotype. TEM analysis of negative-stained thin sections should be sufficient to support the light microscopic evidence already presented, as it should allow visualisation of the (presumably) membrane-bound multiple digestive vacuoles proposed to be produced by the treated parasites. Please also rephrase the rather ungrammatical subtitle near the top of page 8.

3. Although the authors put considerable effort into demonstrating that treatment of parasites with compound 3 produces a similar phenotype to that of gene knockdown, I was surprised to note that they did not attempt to examine whether parasites genetically depleted of PfA-M17 demonstrate increased sensitivity to compound 3 (similar to that observed when grown in medium lacking amino acids other than isoleucine). This is a commonly-used approach to demonstrate on-target activity of small compounds; it would make good use of their knockdown mutant and would certainly add support to their assertions. The authors should carefully titrate the potency of compound 3 against wild-type parasites, PfA-M17-HAglmS parasites and PfA-M17-HAglmS parasites treated with sub-lethal levels of GlcN in order to test whether knockdown of PfA-M17 leads to enhanced sensitivity to the inhibitor.

4. In Figure 8B, two peptides indicated by green dots are claimed to have been confirmed by MS/MS fragmentation to be likely derived from haemoglobin. What are the sequences of these peptides? Are they Glu-Glu-Glu-Lys-Trp and Asp-Phe-Ile-Tyr-Tyr indicated by black bars in Figure 8A? This wasn't clear to me. If so, in the proposed parent haemoglobin sequence, are these peptides preceded by residues (e.g. Leu or Trp) known to be preferred substrates for PfA-M17? Please discuss this issue in the Discussion.

Additional textual recommendations:

In my view, although much of the manuscript is well written and clear, detailed assessment of the entire text should be carried out in order to improve clarity and presentation. Some parts of the manuscript could be much better written. These include the Abstract, which includes the slightly misleading comment '…these parasites are now resistant to all anti-malaria drugs used throughout the world…' (true for only rare *P. falciparum* isolates, rather than all isolates as implied). The structure of the Abstract could also be improved by re-ordering some of the text; as it stands, the veracity of the penultimate sentence relies on the compound 3 being on-target, so to start the last sentence with 'We further show that…' does not properly convey the logic of the author's arguments; this point should be made earlier in the paragraph. I would strongly suggest that the paper would be much improved by going through all the text in similar detail, ensuring the flow of logic is optimal. Please replace all examples of the word 'utilize' with 'use', which means exactly the same but is shorter. Also please number all lines in the text, as this makes it much easier for the reviewer to refer to specific parts of the text.

Alternative, previously-used names for the two aminopeptidases that are the subject of this work should be mentioned in the Introduction (e.g. PfM17LAP and PfM1AAP) to improve clarity. Also, the penultimate paragraph of the Introduction should include a clear statement of what is known of the substrate specificity of PfA-M17 and PfA-M1 – i.e. that both enzymes are thought to cleave single residues from the N-terminal end of short peptides with free amino termini. This is mentioned on page 9, but would usefully come earlier in the manuscript.

There is a nearly 20-fold difference between the K_i_ of compound 3 against recombinant PfA-M17 (~18 nM) and the EC50 of compound 3 in parasite growth assays (~326 nM). This may well be due to stability or membrane-permeability issues, but the authors should at least comment on this. Obviously, the work would be enormously strengthened by a demonstration that compound 3 is effective at clearing parasitaemia in an appropriate in vivo animal model (e.g. a humanised rodent *P. falciparum* model), but this would require a lead compound with suitable PK/PD or ADEME properties and is not essential for the thrust of this paper.

*Reviewer #2 (Recommendations for the authors):*

In this manuscript, Edgar et al. report a two-pronged characterization of an metallo-aminopeptidase of the M17 family (named PfA-M17, also referred to in the literature as a "leucine aminopeptidase" due to its preference for non-polar N-terminal residues). This is one of two metallo-aminopeptidases, the other originating from the M1 family, that have been implicated in general peptide catabolism in the parasite. Most of the peptide substrates are assumed to derive from host erythrocyte hemoglobin, which is endocytosed in massive quantities and degraded in the digestive vacuole. While there is substantial evidence for M1 aminopeptidase activity in the digestive vacuole, it has been proposed that at least some globin peptides are transported out of the vacuole for hydrolysis in the cytosol, and cytosolic PfA-M17 is likely a key enzyme in the generation of cytosolic amino acids. Both of these aminopeptidases are thought to be essential in *Plasmodium falciparum*, although much of the evidence for this derives from the inability to generate viable parasites with disrupted genes. Here, the authors address this gap in knowledge using genetic and pharmacologic approaches.

In the first part of the manuscript, the authors generate a parasite line carrying a modified PfA-M17 gene to enable an inducible knockdown. For the first time, they are able to follow the effects of depletion of PfA-M17 activity on parasite growth. They demonstrate that PfA-M17 knockdown is lethal and convincingly establish the essentiality of the enzyme for asexual replication in the erythrocyte, thus confirming with direct evidence what has long been suspected. An interesting aspect of these studies is that growth of the knockdown parasites take two replication cycles to fully stall, possibly due to the need to deplete PfA-M17 that exists at the time of initiation of knockdown.

The authors notice an intriguing effect of PfA-M17 knockdown: an apparent proliferation of hemozoin crystals in the parasite. In wild-type *P. falciparum*, hemozoin (crystalline heme from hemoglobin metabolism) accumulates in a single, large digestive vacuole. The authors' observations suggest a substantial disruption of either endocytic processes or vacuolar function. While this is an interesting finding, some caution is warranted in referring to these separate hemozoin crystals as "multiple digestive vacuoles". It is not possible to determine from Giemsa smears whether the authors' observations reflect multiple vacuoles or loss of integrity of a single vacuole, resulting in free hemozoin in the cytosol. One way to resolve this issue would be to demonstrate separate, intact vacuoles using a lysosomotropic dye, such as lysotracker.

The authors then switch gears and describe the development of an inhibitor (termed "3") that is highly selective *in vitro* for PfA-M17 over the M1 family enzyme PfA-M1. The inhibitor has a hydroxamate pharmacophore that interacts with the catalytic zinc ions in metallo-aminopeptidases, and is an extension of previous work on this chemical family by McGowan and colleagues. The inhibitor is highly potent against PfA-M17 *in vitro* (the inhibition constant is 18 nM) with over 100-fold selectivity for PfA-M17 over PfA-M1. A co-crystal structure is provided, which illustrates that the enzyme accommodates the inhibitor in an expansive S1 subsite, as has been previously shown for structurally related inhibitors. The authors then test the anti-malarial activity of inhibitor 3 and find that it kills parasites with an EC50 value of 330 nM. Interestingly, inhibitor treatment leads to a pattern of hemozoin fragmentation similar to that observed with the PfA-M17 knockdown.

A major challenge in chemical biology is establishing the in-cell specificity of a novel chemical tool. One of the weaknesses of this part of the paper is that the specificity of inhibitor 3 is not convincingly demonstrated. The authors make an attempt in this direction by treating PfA-M17 knockdown parasites with the inhibitor "following knockdown at the point where the growth delay becomes apparent in GlcN treated parasites". If understood correctly, the inhibitor is added when the knockdown parasites are already on their way to dying, and it is unclear how the loss of viability with or without inhibitor says anything about target specificity. Generally well accepted approaches to validating specificity include identifying resistance mutations in the target, overexpressing the target and observing an increase in EC50, or generating a chemical probe for direct labeling.

The authors then conduct a comparative metabolomics analysis of the effects of PfA-M17 knockdown and (separately) inhibitor 3 treatment. The main finding is that, in both cases, changes in peptide levels predominate. While there are some commonalities in the changes in peptide species observed in the two conditions, overall, the heat maps in Figure 7B suggest quite disparate effects. The authors identify 80 out of 149 peptides that are "dysregulated in abundance" in both conditions; these are presented in Figure 8. Although the authors don't explicitly claim that these peptides are substrates of knocked -down or inhibited PfA-M17, that seems to be the implication. However, inspection of the peptide sequences suggests that the vast majority of dysregulated peptides are unlikely to be PfA-M17 substrates. This is because PfA-M17 has a rather strict specificity, described by the authors as "a strong preference for a P1 leucine and tryptophan residue". Yet only 2 out of 80 peptides possess P1 Leu or Trp, whereas many contain highly polar N-terminal residues (Glu, Gln, Asp, Asn, etc) and are clearly not PfA-M17 substrates. This suggests that the disruption of PfA-M17 is having knock-on effects that cause perturbation of peptide catabolism on a much wider scale, perhaps due to the loss of vacuolar integrity. In addition, it is noted that the experiments are conducted at 10x the EC50, or 3.3 uM. This could result in significant inhibition of PfA-M1, which has an *in vitro* K_i_ of ~4 uM. The obvious incompatibility of the peptides in Figure 6 with the established specificity of PfA-M17, and the possibility of cross-inhibition of PfA-M1, should be discussed in the paper.

Lastly, the authors implicate PfA-M17 in amino acid production from hemoglobin by culturing parasites in medium lacking all amino acids but isoleucine, which is not present in human hemoglobin. The idea is that this sensitizes parasites to the diminution of amino acid production from hemoglobin peptides. The authors present data in Figure 9 that suggests that inhibitor 3, but not artemisinin, is more potent when parasites are grown in amino acid-restricted media. These claims are undercut by a technical issue, namely that the inhibitor concentrations used in Figure 9 do not permit the establishment of an upper baseline for the inhibition curves, compromising their reliability. This issue is especially acute for the curves conducted in amino acid restricted media, which do not reach 100% growth.

Please add a scale bar to Figure 2A.

Figure 2C, upper panel, is cropped in an odd manner. It should be cropped such that the band in interest is in the center of the panel.

Supplemental Figure S1 is provided to validate the specificity of the anti-PfAM17 antibody used to generate Figure 2C. However, it looks quite different from the source data file provided (Figure 2-source data 5) which shows at least one major non-specific band (unless this is an alternate form of PfAM17). In the spirit of transparency, it is recommended that this source data file be included in Figure S1.

Also related to Supplemental Figure S1: this presents data from two separate images (pre-bleed and final bleed) that are merged to give the impression of a single image. This is contrary to best practice; specifically, images that derive from separate blots or exposures should be clearly indicated as such with an intervening space.

"Knockdown of PfA-M17…but not the quantity of hemozoin". This statement seems to be missing a modifier for "quantity of hemozoin".

"On analysis of PfA-M17-HAglmS parasites the cycle following knockdown…" This presumably refers to cycle 2 as depicted in Figure 3. This should be explicitly stated to remove any ambiguity.

"the main chain amine of Gly489" should presumably read amide.

"Using the structure of 4ZX4.pdf as a template…." What is this structure?

In Figure 5C, it is not clear what "%growth" refers to. If this is a SYBR green assay, it is sensitive to nucleic acid content, which is not necessarily the same thing as "growth".

All of the supplemental figures are at low resolution resulting in a highly pixelated appearance which compromises legibility. Hi-res versions should be provided.

*Reviewer #3 (Recommendations for the authors):*

This manuscript offers further characterization of PfA-M17, a *P. falciparum* aminopeptidase that has been studied for some years, and was previously shown to be an essential protein predicted to function in late steps of hemoglobin hydrolysis by erythrocytic parasites. The new report adds valuable demonstration of impacts of PfA-M17 knockdown, description of the synthesis and characterization of a novel PfA-M17 inhibitor with high nanomolar activity against cultured parasites, and a metabolomic analysis of inhibited parasites adding evidence for hemoglobin to be the primary target of the protease. The manuscript is well-written, the described experiments were appropriate, and the conclusions are generally convincing. The manuscript adds importantly to our understanding of the roles of aminopeptidases in parasite biology. However, there are some important concerns, as discussed below.

1) Abstract. The statement "these parasites are now resistant to all anti-malaria drugs" is wildly misleading. It is true that resistance is seen to many (but not all; consider lumefantrine and pyronaridine) available antimalarial drugs, but the vast majority of infections are readily treatable with available ACTs. This statement implies otherwise, and it should be changed. There clearly is a need for new antimalarials, but misleading hyperbole to make this claim is unnecessary.

2) Introduction, first paragraph. The WHO dramatically increased estimates for malaria deaths, including revision of past estimates. See the 2021 World Malaria Report. The paragraph should more clearly indicate the role of ACTs in treating falciparum malaria. Artemisinin resistance is NOT "widespread" throughout Asia, but rather to the best of our knowledge confined to a quite small percentage of this large continent, the Greater Mekong sub-region. Limited evidence for spread to India is controversial. Many other parts of Asia have malaria, but without evidence for artemisinin resistance. The statement that artemisinin resistance has recently spread to PNG is both misleading and incomplete. All that is published on PNG is a report on a few isolates with worrisome K13 mutations; it is not clear if this means that resistance is seen in PNG. On the other hand, recent results from Rwanda and Uganda are more convincing, and more worrisome, for spread of artemisinin resistance, and the introduction should better (albeit briefly) summarize the newest data.

3) Introduction. "PfA-M17 almost exclusively cleaves leucine and tryptophan *in vitro*" is a bit too simplistic. It would be better for the authors to more accurately, but briefly describe the activity of an aminopeptidase, which is to cleave AAs from the amino-terminal end of peptides or proteins. Indeed, this is provided, but in the middle of the Results section rather than in the Introduction, where it will be more useful for readers.

4) Results, first paragraph. The relevance of this discrepancy is unclear, but it is misleading to state that the HA tagged PfA-M17 construct was seen on a Western blot at "slightly lower than the predicted 72 kDa size". In fact the size appears to be much lower than 72 kDa, and the authors should discuss potential explanations for this surprising result.

5) Results, P. 8. The following subtitle has uncertain meaning, and should be rewritten: "Knockdown of PfA-M17 results in multiple digestive vacuoles forming during trophozoite stage but not the quantity of hemozoin".

6) Figure 4A. The demonstration of multiple DVs is interesting and potentially very important in terms of characterizing mechanisms, but the micrograph showing this phenotype is sub-optimal. A larger, sharper image, and ideally multiple images showing this phenotype will be helpful. The figure suggests that only a small minority of parasites had >1 DV; quantification of this result (what percentage of control and treated parasites had >1 DV?) would be helpful.

7) Results, P. 12-13. The metabolomics experiments are elegant, but the argument derived from these studies is somewhat unsatisfying. Inhibition of an aminopeptidase is expected to result in accumulation of peptides. But, after knockdown or treatment with an inhibitor of the aminopeptidase, is this accumulation specifically of Hb-derived peptides? The results are suggestive, but the peptides shown to accumulate were all short (2-5 AAs), so identification as Hb-derived is uncertain; many might similarly be seen as hydrolysis products of any large protein. Can the authors better justify their argument, perhaps with a simple mathematical explanation for why assignment of this number of short peptides as Hb products is strongly suggestive of Hb peptides as the natural targets of PfA-M17? In other words, how sure can we be that assignment of a number of short peptides to hemoglobin is not simply due to chance?

8) P. 14. Artemisinin should not be capitalized.

9) Discussion. The discussion offers a nice discussion of the relevance of different substrate specificities of different plasmodial aminopeptidases. The report offers a detailed list of peptides that accumulate with knockdown or inhibition of PfA-M17. Are these peptides consistent with loss of the specific proteolytic specificity of PfA-M17 (with preference for P1 leucine and tryptophan residues)?

10) Discussion. Some additional discussion should address the presentation of PfA-M17 as a potential drug candidate. What is the likelihood that improved potency can be engineered, as the current lead offers relatively modest (high nanomolar) *in vitro* potency? What is the evidence that compound 3 does not inhibit host proteases? Are there concerns about the quite slow action of the inhibitor against cultured parasites?

[Editors’ note: further revisions were suggested prior to acceptance, as described below.]

Thank you for resubmitting your work entitled "Genetic and chemical validation of *Plasmodium falciparum* aminopeptidase *PfA*-M17 as a drug target in the hemoglobin digestion pathway" for further consideration by *eLife*. Your revised article has been evaluated by Dominique Soldati-Favre (Senior Editor) and a Reviewing Editor.

The manuscript has been improved but there are some remaining issues that need to be addressed, as outlined below:

Both reviewers concur that the TPP data need to be clarified and these revisions should be made. No new experimentation should be required for that.

*Reviewer #1 (Recommendations for the authors):*

The authors have substantially revised the manuscript in response to the prior review and as a result the claims in the manuscript are more robust. The authors have provided an impressive quantity of new data; notable examples include the analysis of fragmented food vacuoles by TEM, MS/MS analysis of key peptide metabolites in the metabolomics experiment and analysis of their probable origin from globin, and attempts to establish the specificity of compound 3 by a variety of approaches. Overall, this is a very solid manuscript that convincingly establishes the essentiality of PfA-M17 and its role in peptide catabolism.

The only remaining concern relates to the newly-added thermal proteomics profiling (TPP) study. In this approach, addition of a small molecule ligand to a proteome is expected to specifically stabilize the target against thermal denaturation. The authors conducted TPP at two concentrations of compound 3 and present these data in Figure 6. They claim that "only a single protein was significantly stabilized across the two drug concentrations…PfA-M17". However, looking at Figure 6A, it is clear that there are at least a dozen proteins that are more highly stabilized in both drug concentrations. Is the implication that none of these proteins was "significantly stabilized"? If so, it doesn't make sense on an intuitive level that all of the proteins in that exhibit a higher fold-change are not significant, whereas PfA-M17, which at the edge of the undifferentiated mass of "unchanged" proteins, is significant. To allow readers more insight into this study, the authors should provide (as supplemental data) the identities of the proteins in the upper right quadrant of the 3 µM fold change vs. 12 µM fold change plot (about 15-20 proteins), along with relevant mass spec data (number of unique peptides, #s of technical replicates in which they are observed) and the fold-change values and associated statistics. It would also be helpful to include the data for PfA-M1 as this is the most likely off-target. This would help to put the claims of on-target binding of compound 3 on a solid foundation.

The authors should also provide more details on the "LC-MS/MS and data analysis" methods related to TPP. The statement "Following protein identification and quantification…" needs to be elaborated on, as this is the most important set of experimental details related to understanding Figure 6. Things that would be useful to know: was there a minimum unique peptide cutoff? How many peptides from a given protein were used in quantitation? Did these have to appear in all technical replicates? Data in Figure 6B are reported as "Protein relative abundance"; relative to what?

*Reviewer #2 (Recommendations for the authors):*

In this revised version of their manuscript, the authors have dealt satisfactorily with the majority of the issues raised in response to the original submission. In particular, the use of TEM to better visualise the 'multiple DV' phenotype resulting from PfA-M17 knockdown adds significantly to the study, and the thermal proteomic profiling (TPP) analysis helps substantiate Pf-M17 as an in vivo target of compound 3. These additions to the work are appreciated and definitely improve the manuscript. However, there is still a lack of clarity in some sections of the manuscript, generally due to inaccurate language.

Requested modifications:

Lines 309-311 and Figure 6 – This part of the manuscript describes the TPP analysis. It is stated in the text that '…only a single protein was significantly stabilised across the two drug concentrations…'. Furthermore, a similar statement is made in the Figure 6 legend, where it is stated that: '…a single protein (Pf3D7_1446200) was significantly altered at both concentrations…'. However, the volcano plots shown in Figure 6 show that in fact several proteins (~12 and ~14 respectively) fall into the right-hand top quadrant of the plots at drug concentrations of 3 μM and 12 μM, suggesting that these are stabilised by drug treatment too. Do the authors mean that PfA-M17 was the only protein that was in common between these two protein subsets showing increased stability at the two drug concentrations used? If so, they should specifically and more accurately state this.

Lines 1222 and 1228 (Figure 7 legend) – replace 'medium' with 'median'? Alternatively, explain these terms better.

Supplementary Figure 2A and 2B: Please indicate the concentration of glucosamine used in both the figure legend and the figure panels.

Supplementary Figure 3: these data were included in response to the suggestion to include a full microscopic field of the control and glucosamine-treated Pf-M17-HAglmS parasites, in order to demonstrate depletion of the HA-tagged protein at the single parasite level following treatment with glucosamine. Unfortunately, the provided figure is still sub-optimal; it should include panels in which the same parasites are probed with a control antibody (i.e. specific for an irrelevant parasite protein), as well as DIC or bright-field images to show the morphology of the parasites. Please provide a suitable replacement image.

---

## [Author Response]

[Editors’ note: the authors resubmitted a revised version of the paper for consideration. What follows is the authors’ response to the first round of review.]

Comments to the Authors:A decision has been reached after consultation between the reviewers. Based on these discussions and the individual reviews below, the reviewers concur that the work offers a sizable contribution to our understanding of the roles of aminopeptidases in parasite biology using genetic and pharmacologic approaches. However, they also shared some concerns and identified weaknesses that are summarized below. Some of the major points (1 and 3) as well as the minor issues raised in individual reviews need to be addressed experimentally prior to consideration for acceptance at eLife.

We thank the editor and the reviewers for recognising the contribution our work makes to the understanding of the roles of aminopeptidases in parasite biology. The reviewers have made some important points and we have taken this feedback on board and carried out additional experimentation to address the outstanding concerns, particularly those centred around validating the target of compound 3 and documenting the digestive vacuole phenotype more thoroughly by TEM.

1. The morphological phenotype observed both in PfA-M17 knockdown and in wt parasites during inhibitor treatment is interesting but the micrograph showing this phenotype is sub-optimal and needs to be consolidated. The authors should be inspired by reviewer suggestions to use lysosomotropic dye, such as lysotracker or to document the phenotype by electron microscopy.

We thank the reviewers for their helpful suggestions in improving the manuscript. We have now conducted transmission electron microscopy to examine this interesting phenotype and demonstrate that in contrast to parasites treated with vehicle control, in which a single digestive vacuole (DV) is observed, parasites treated with compound 3 exhibit multiple membrane-bound digestive vacuoles containing hemozoin. This new data is now provided in Figure 7. We have also quantified the number of parasites with multiple hemozoin crystals, and this data is also provided in Figure 7. A discussion of these results can be found in the Section ‘Loss of PfA-M17 results in the formation of multiple digestive vacuoles’ (from line 312) and in the Discussion section (from line 536).

2. The demonstration that a selective inhibitor of PfA-M17 kills the parasite *in vitro* lacks novelty, as similar findings have been published previously. Also, the co-crystal structure which illustrates that the enzyme accommodates the inhibitor in an expansive S1 subsite, has been previously shown for structurally related inhibitors.

The novel and new discovery we report in this manuscript is not the M17 selective inhibitor, known as compound 3, rather it’s the characterisation of the essentiality and function of PfA-M17. However, this compound is an excellent additional tool to provide chemical validation of the PfA-M17 target in parallel to our genetic approach, whilst showing modest effect against parasites. Altogether, this provides excellent validation of PfA-M17 as a potential novel drug target. Previous work did use a bestatin-derived activity-based probe; however, this molecule lacked the selectivity over PfA-M1 compared to 3 and failed to identify key morphological changes that occur in the parasite when PfA-M17 activity is inhibited (Harbut et al., 2011). In our current study, we were able to clearly identify the function of PfA-M17 using both a genetic and chemical induced reduction of PfA-M17 activity in the parasite.

The inclusion of the structural data validates the competitive binding mechanism of action of 3 and was not meant to be a centre piece of this study. It is included for transparency with regard to the complete characterisation of 3 (synthesis and activity) which has not been reported before and is the most selective PfA-M17 inhibitor produced to date.

3. Importantly the data addressing the specificity of inhibitor 3 is lacking. The rationale for adding the inhibitor when the knockdown parasites are already on their way to dying to demonstrate specificity is not convincing. Suggested approaches to validating specificity include identifying resistance mutations in the target, overexpressing the target and observing an increase in EC50, observing increased sensitivity to chemical inhibition following genetic knockdown, or generating a chemical probe for direct labeling.

Since an activity-based probe based on the bestatin scaffold designed to inhibit PfA-M17 led to ring-stage arrest and the phenotype of our knockdown did not occur until trophozoite stage, we had originally attempted to demonstrate that inhibitor 3 was on target by assessing the EC_50_ against knockdown parasites in sub-lethal concentrations of 3. However, we found that we were unable to demonstrate 100% growth of any controls treated with DMSO, which we attributed to the delay in death seen from the knockdown alone. We made corrections to ‘Analysis of parasites depleted of PfA-M17 alongside treatment with the specific PfA-M17 inhibitor’ to reflect that the compound was actually added just after reinvasion the cycle following knockdown before the phenotype becomes apparent at the trophozoite stage in order to show that there is no earlier killing of parasites which could be attributed to off target effects.

Accordingly, and based on the reviewers’ feedback, we tried two approaches to demonstrate compound 3 specificity. Firstly, we attempted to generate compound 3 resistant parasites for downstream genomic analysis of mutations, but we were unable to recover any parasites from multiple dishes after 4 months of culturing using the published method whereby a high number of parasites are treated with 3x EC_90_ (Okombo et al., 2021). We were, however, able to confirm PfA-M17 as the target using another approach, that being unbiased thermal proteomics profiling (TPP). Here, using two separate drug concentrations, we identified only a single protein, that being PfA-M17, to be stabilised, demonstrating that compound 3 targets PfA-M17. These results can be found under the heading ‘Thermal Proteomics Profiling (TPP) confirmed PfA-M17 to be the target of compound 3’ (from line 294) and are also presented in Figure 6.

4. The comparative metabolomics analysis leads to confounding results.Given the strict specificity PfA-M17 (strong preference for a P1 leucine and tryptophan residue), the accumulating peptides are not consistent with loss of the specific proteolytic specificity of PfA-M17. Moreover, the comparison of the impact of PfA-M17 knockdown or compound 3 treatment show modest overlap. It is plausible that a wide perturbation of catabolism would result from a loss of vacuolar integrity.

Indeed, we agree that the broad impact of PfA-M17 depletion/inhibition is somewhat surprising, as it is not merely limited to Leu/Trp peptides. Whilst it is possible that PfA-M17 has a broader substrate specificity within the intracellular context than what has been demonstrated in simplified enzyme assays, the extensive dysregulation (~80 peptides) observed here suggests a secondary impact on proteolytic processes. The effect may be due to direct inhibition of other proteases/peptidases by the accumulation of PfA-M17 substrates, dysfunctional digestive vacuole activity associated with the observed morphology, and/or secondary dysregulation of proteolytic enzyme levels or functions. Some comments about these potential effects have been added to the discussion from lines 484.

The claim of ‘modest overlap’ is not supported by the data. PfA-M17 knockdown induced accumulation of 77 of the same 80 peptides that were also accumulated following 3 treatment. This is an incredibly high overlap, especially considering that accumulation of these peptides has not been observed for over 100 unrelated compounds that have been tested with this metabolomics methodology (Creek et al., 2016, Birrell et al., 2020, Giannangelo et al., 2020). The text in the Results section ‘PfA-M17 plays a role in the degradation of hemoglobin-derived peptides’ (from line 357) has been edited to clarify the extensive similarity of these profiles, and a heatmap with hierarchical clustering included in Figure 9 to demonstrate unbiased clustering of these samples.

5. The authors should discuss the risk of inhibition of host proteases and the potential limitation of the slow action of the inhibitor against cultured parasites.

This is a valid comment, and this has now been included in the discussion from line 514. The slow action of 3 is not strictly true, and while Giemsa-images show a delay in parasite morphological growth, washout experiments show that both 24 or 48 hour treatments have a significant effect on parasite survival, with this additionally showing that the compound is most active against trophozoite stage parasites, in keeping with the knockdown phenotype.

Reviewer #1 (Recommendations for the authors):There is a need to discover new targets for new kinds of drugs to tackle malaria, which is a major health threat across the globe. In this work, the authors took 3 complementary approaches to investigate the function, essentiality and druggability of an enzyme of the malaria parasite *Plasmodium falciparum*, an aminopeptidase called PfA-M17. First they used a conditional gene knockdown strategy to show that parasites depleted of PfA-M17 cannot proliferate in culture. Next, they designed a small molecule compound designed to inhibit PfA-M17 and showed that this compound (called compound 3) both inhibited recombinant PfA-M17 with sub-μM potency and killed the parasite, producing a phenotype similar to that resulting from gene knockdown. In efforts to demonstrate on-target efficacy of compound 3, the authors additionally showed that parasites depleted of PfA-M17 by gene knockdown are killed by compound 3 with similar kinetics (and with a similar phenotype) to parasites expressing wild-type levels of the enzyme. Finally, by comparing the metabolomic profile of gene knockdown and compound 3-treated parasites, the authors showed that a number of peptide products likely derived from haemoglobin accumulated in both parasite populations. Collectively these results support the on-target activity of compound 3 and provide evidence that PfA-M17 acts to degrade haemoglobin-derived peptides in vivo. The authors conclude that PfA-M17 is a potential target for a new type of antimalarial therapy based on inhibitors of the enzyme.Strengths: Overall, the work is well-presented and uses an appropriate combination of orthogonal technologies to interrogate the function of this intriguing enzyme. The study makes a useful contribution to knowledge of basic biology of the parasite, but importantly also provides insights valuable for drug discovery efforts, important in the light of growing evidence for emerging artemisinin resistance in Africa. The phenotypic response to enzyme knockdown (genetic or chemical) by the apparent generation of multiple digestive vacuoles is intriguing and potentially very informative, but needs to be better supported experimentally.Weaknesses: Whilst the majority of the conclusions are very well supported by the experimental data, a major aspect of the work – the demonstration that a selective inhibitor of PfA-M17 kills the parasite *in vitro* – lacks novelty, as similar findings have been published previously both by some of these authors and by others (e.g. PMID: 27449897, PMID: 26406322, PMID: 26807544). The conclusion that genetic knockdown of PfA-M18 or inhibition of the enzyme with compound 3 results in multiple digestive vacuoles would also be much better supported by electron microscopic analysis rather than only light microscopy as in the current manuscript.Specific issues to be addressed experimentally:1. The knockdown data provided in Figure 3 are good, but it would be more convincing to see these data supplemented by immunofluorescence data (ideally showing a full field of parasites) probing with anti-HA antibodies or anti-PfA-M17 antibodies. These data would be complementary to the western blot results in that they show expression levels in individual parasites.

We have included a full field image of Pf-M17-HAglmS parasites in cycle 2 after glucosamine addition and probed with anti-HA antibodies alongside DAPI, where upon merging of the images we see little to no expression of HA, and thus PfA-M17 in parasites. This has now been included as Supplementary Figure 3 and commented on in the Results section under ‘Knockdown of PfA-M17 expression reveals its essentiality to parasite survival’ at line 169’.

2. The observation that genetic knockdown of PfA-M17 or treatment with compound 3 results in parasites exhibiting multiple haemozoin granules is very interesting indeed and may indeed shed light on the parasite's response to these perturbations, as speculated by the authors in the Discussion. However, in my view these light microscopic observations need to be supported by electron microscopic analysis in order to provide a definitive evaluation of this phenotype. TEM analysis of negative-stained thin sections should be sufficient to support the light microscopic evidence already presented, as it should allow visualisation of the (presumably) membrane-bound multiple digestive vacuoles proposed to be produced by the treated parasites. Please also rephrase the rather ungrammatical subtitle near the top of page 8.

On reflection we agree that it is not possible to discern from light microscopy whether the multiple hemozoin granules represent multiple digestive vacuoles or loss of integrity of a single vacuole leading to release of free hemozoin into the cytosol and have edited the results to reflect this. To examine this phenotype further, we have examined parasites by transmission electron microscopy (TEM). As multiple hemozoin crystals could been seen both after PfA-M17 knockdown and after treatment with 3 and TPP confirmed PfA-M17 is the target of 3, the TEM was performed on 3-treated parasites. Accordingly, for the TEM, parasites were treated with 10xEC_50_ of 3 or with the same concentration of DMSO (vehicle control) and at a developmentally similar age, once all parasites contained hemozoin, parasites were fixed and imaged. Analysis of these images shows parasites treated with 3 contain multiple membrane-bound digestive vacuoles containing hemozoin. In contrast, DMSO-treated parasites displayed a single digestive vacuole containing hemozoin that is comparable to other published images of digestive vacuoles in wild type parasites. As we used the compound for treatment and imaging by TEM, we have combined this data into Figure 7 which shows both the hemozoin fragment counts, the TEM images and the hemozoin quantification. Discussion of these results is now can be found under the edited section title ‘Loss of PfA-M17 results in the formation of multiple digestive vacuoles’ (from line 312).

3. Although the authors put considerable effort into demonstrating that treatment of parasites with compound 3 produces a similar phenotype to that of gene knockdown, I was surprised to note that they did not attempt to examine whether parasites genetically depleted of PfA-M17 demonstrate increased sensitivity to compound 3 (similar to that observed when grown in medium lacking amino acids other than isoleucine). This is a commonly-used approach to demonstrate on-target activity of small compounds; it would make good use of their knockdown mutant and would certainly add support to their assertions. The authors should carefully titrate the potency of compound 3 against wild-type parasites, PfA-M17-HAglmS parasites and PfA-M17-HAglmS parasites treated with sub-lethal levels of GlcN in order to test whether knockdown of PfA-M17 leads to enhanced sensitivity to the inhibitor.

Since an activity-based probe based on the bestatin scaffold designed to inhibit PfA-M17 led to ring-stage arrest and the phenotype of our knockdown did not occur until trophozoite stage, we had originally attempted to demonstrate that inhibitor 3 was on target by assessing the EC_50_ against knockdown parasites in sub-lethal concentrations of 3. However, we found that we were unable to demonstrate 100% growth of any controls treated with DMSO, which we attributed to the delay in death seen from the knockdown alone. We made corrections to ‘Analysis of parasites depleted of PfA-M17 alongside treatment with the specific PfA-M17 inhibitor’ to reflect that the compound was actually added just after reinvasion the cycle following knockdown before the phenotype becomes apparent at the trophozoite stage in order to show that there is no earlier killing of parasites which could be attributed to off target effects. We have also now determined the target of 3 to be PfA-M17 using the unbiased thermal proteomic profiling (TPP) approach.

4. In Figure 8B, two peptides indicated by green dots are claimed to have been confirmed by MS/MS fragmentation to be likely derived from haemoglobin. What are the sequences of these peptides? Are they Glu-Glu-Glu-Lys-Trp and Asp-Phe-Ile-Tyr-Tyr indicated by black bars in Figure 8A? This wasn't clear to me. If so, in the proposed parent haemoglobin sequence, are these peptides preceded by residues (e.g. Leu or Trp) known to be preferred substrates for PfA-M17? Please discuss this issue in the Discussion.

Apologies for the lack of clarity. The black bar in figure 8A was indicative of those peptides that only increased after 3 treatment and not after knockdown – we have now indicated this more clearly in the figure legend. We have increased the MS/MS data to include more sequences that are confirmed to be likely derived from hemoglobin and have additionally identified sequences that cannot originate from hemoglobin, which is indicated now in Figure 9B. Please refer to the revised section ‘PfA-M17 plays a role in the degradation of hemoglobin-derived peptides’ from line 357 and the discussion on the mass-spec results at 436-444 and from line 484.

Additional textual recommendations:In my view, although much of the manuscript is well written and clear, detailed assessment of the entire text should be carried out in order to improve clarity and presentation. Some parts of the manuscript could be much better written. These include the Abstract, which includes the slightly misleading comment '…these parasites are now resistant to all anti-malaria drugs used throughout the world…' (true for only rare *P. falciparum* isolates, rather than all isolates as implied). The structure of the Abstract could also be improved by re-ordering some of the text; as it stands, the veracity of the penultimate sentence relies on the compound 3 being on-target, so to start the last sentence with 'We further show that…' does not properly convey the logic of the author's arguments; this point should be made earlier in the paragraph. I would strongly suggest that the paper would be much improved by going through all the text in similar detail, ensuring the flow of logic is optimal.

We thank the reviewer for this helpful feedback and have taken their suggestions on board and revised the abstract and manuscript accordingly.

Please replace all examples of the word 'utilize' with 'use', which means exactly the same but is shorter.

Corrected.

Also please number all lines in the text, as this makes it much easier for the reviewer to refer to specific parts of the text.

Line numbers have now been added to the text.

Alternative, previously-used names for the two aminopeptidases that are the subject of this work should be mentioned in the Introduction (e.g. PfM17LAP and PfM1AAP) to improve clarity.

We agree this is important and this has now been added to Introduction.

Also, the penultimate paragraph of the Introduction should include a clear statement of what is known of the substrate specificity of PfA-M17 and PfA-M1 – i.e. that both enzymes are thought to cleave single residues from the N-terminal end of short peptides with free amino termini. This is mentioned on page 9, but would usefully come earlier in the manuscript.

We agree that this would be useful to the reader and have now mentioned what the substrate specificities of the aminopeptidases are at the penultimate paragraph of the introduction.

There is a nearly 20-fold difference between the K_i_ of compound 3 against recombinant PfA-M17 (~18 nM) and the EC50 of compound 3 in parasite growth assays (~326 nM). This may well be due to stability or membrane-permeability issues, but the authors should at least comment on this.

We now comment on this in the discussion from line 499.

Obviously, the work would be enormously strengthened by a demonstration that compound 3 is effective at clearing parasitaemia in an appropriate in vivo animal model (e.g. a humanised rodent *P. falciparum* model), but this would require a lead compound with suitable PK/PD or ADEME properties and is not essential for the thrust of this paper.

We agree that this would add important information to the druggability of PfA-M17 alone but believe it is outside the scope and purpose of this study.

Reviewer #2 (Recommendations for the authors):In this manuscript, Edgar et al. report a two-pronged characterization of an metallo-aminopeptidase of the M17 family (named PfA-M17, also referred to in the literature as a "leucine aminopeptidase" due to its preference for non-polar N-terminal residues). This is one of two metallo-aminopeptidases, the other originating from the M1 family, that have been implicated in general peptide catabolism in the parasite. Most of the peptide substrates are assumed to derive from host erythrocyte hemoglobin, which is endocytosed in massive quantities and degraded in the digestive vacuole. While there is substantial evidence for M1 aminopeptidase activity in the digestive vacuole, it has been proposed that at least some globin peptides are transported out of the vacuole for hydrolysis in the cytosol, and cytosolic PfA-M17 is likely a key enzyme in the generation of cytosolic amino acids. Both of these aminopeptidases are thought to be essential in *Plasmodium falciparum*, although much of the evidence for this derives from the inability to generate viable parasites with disrupted genes. Here, the authors address this gap in knowledge using genetic and pharmacologic approaches.In the first part of the manuscript, the authors generate a parasite line carrying a modified PfA-M17 gene to enable an inducible knockdown. For the first time, they are able to follow the effects of depletion of PfA-M17 activity on parasite growth. They demonstrate that PfA-M17 knockdown is lethal and convincingly establish the essentiality of the enzyme for asexual replication in the erythrocyte, thus confirming with direct evidence what has long been suspected. An interesting aspect of these studies is that growth of the knockdown parasites take two replication cycles to fully stall, possibly due to the need to deplete PfA-M17 that exists at the time of initiation of knockdown.The authors notice an intriguing effect of PfA-M17 knockdown: an apparent proliferation of hemozoin crystals in the parasite. In wild-type *P. falciparum*, hemozoin (crystalline heme from hemoglobin metabolism) accumulates in a single, large digestive vacuole. The authors' observations suggest a substantial disruption of either endocytic processes or vacuolar function. While this is an interesting finding, some caution is warranted in referring to these separate hemozoin crystals as "multiple digestive vacuoles". It is not possible to determine from Giemsa smears whether the authors' observations reflect multiple vacuoles or loss of integrity of a single vacuole, resulting in free hemozoin in the cytosol. One way to resolve this issue would be to demonstrate separate, intact vacuoles using a lysosomotropic dye, such as lysotracker.

On reflection we agree that it is not possible to discern from light microscopy whether the multiple hemozoin granules represent multiple vacuoles or loss of integrity of a single vacuole leading to release of free hemozoin in the cytosol and have edited the results to reflect this. To examine this phenotype further, we have examined parasites by transmission electron microscopy (TEM). As multiple hemozoin crystals could been seen both after PfA-M17 knockdown and after treatment with 3, the TEM was performed on 3-treated parasites. Accordingly, for the TEM, parasites were treated with 10xEC_50_ of 3 or with the same concentration of DMSO (vehicle control) and at a developmentally similar age, once all parasites contained hemozoin, parasites were fixed and imaged. Analysis of these images shows parasites treated with 3 contain multiple membrane-bound digestive vacuoles containing hemozoin. In contrast, DMSO-treated parasites displayed a single digestive vacuole containing hemozoin that is comparable to other published images of digestive vacuoles in wild type parasites. As we used the compound for treatment and imaging by TEM, we have combined this data into Figure 7 which shows both the hemozoin fragment counts, the TEM images and the hemozoin quantification. Discussion of these results is now can be found under ‘Loss of PfA-M17 results in the formation of multiple digestive vacuoles’.

The authors then switch gears and describe the development of an inhibitor (termed "3") that is highly selective *in vitro* for PfA-M17 over the M1 family enzyme PfA-M1. The inhibitor has a hydroxamate pharmacophore that interacts with the catalytic zinc ions in metallo-aminopeptidases, and is an extension of previous work on this chemical family by McGowan and colleagues. The inhibitor is highly potent against PfA-M17 *in vitro* (the inhibition constant is 18 nM) with over 100-fold selectivity for PfA-M17 over PfA-M1. A co-crystal structure is provided, which illustrates that the enzyme accommodates the inhibitor in an expansive S1 subsite, as has been previously shown for structurally related inhibitors. The authors then test the anti-malarial activity of inhibitor 3 and find that it kills parasites with an EC50 value of 330 nM. Interestingly, inhibitor treatment leads to a pattern of hemozoin fragmentation similar to that observed with the PfA-M17 knockdown.A major challenge in chemical biology is establishing the in-cell specificity of a novel chemical tool. One of the weaknesses of this part of the paper is that the specificity of inhibitor 3 is not convincingly demonstrated. The authors make an attempt in this direction by treating PfA-M17 knockdown parasites with the inhibitor "following knockdown at the point where the growth delay becomes apparent in GlcN treated parasites". If understood correctly, the inhibitor is added when the knockdown parasites are already on their way to dying, and it is unclear how the loss of viability with or without inhibitor says anything about target specificity. Generally well accepted approaches to validating specificity include identifying resistance mutations in the target, overexpressing the target and observing an increase in EC50, or generating a chemical probe for direct labeling.

Since an activity-based probe based on the bestatin scaffold designed to inhibit PfA-M17 led to ring-stage arrest and the phenotype of our knockdown did not occur until trophozoite stage, we had originally attempted to demonstrate that inhibitor 3 was on target by assessing the EC_50_ against knockdown parasites in sub-lethal concentrations of 3. However, we found that we were unable to demonstrate 100% growth of any controls treated with DMSO, which we attributed to the delay in death seen from the knockdown alone. We made corrections to ‘Analysis of parasites depleted of PfA-M17 alongside treatment with the specific PfA-M17 inhibitor’ to reflect that the compound was actually added just after reinvasion the cycle following knockdown before the phenotype becomes apparent at the trophozoite stage in order to show that there is no earlier killing of parasites which could be attributed to off target effects.

Accordingly, and based on the reviewers’ feedback, we tried two approaches to demonstrate compound 3 specificity. Firstly, we attempted to generate compound 3 resistant parasites for downstream genomic analysis of mutations, but we were unable to recover any parasites from multiple dishes after 4 months of culturing using the published method whereby a high number of parasites are treated with 3x EC_90_ (Okombo et al., 2021). We were, however, able to confirm PfA-M17 as the target using another approach, that being unbiased thermal proteomics profiling (TPP). Here, using two separate drug concentrations, we identified only a single protein, that being PfA-M17 to be stabilised, demonstrating that compound 3 targets PfA-M17. These results can be found under the heading ‘Thermal Proteomics Profiling (TPP) confirmed PfA-M17 to be the target of compound 3’ (from line 312) and are also presented in Figure 6.

The authors then conduct a comparative metabolomics analysis of the effects of PfA-M17 knockdown and (separately) inhibitor 3 treatment. The main finding is that, in both cases, changes in peptide levels predominate. While there are some commonalities in the changes in peptide species observed in the two conditions, overall, the heat maps in Figure 7B suggest quite disparate effects. The authors identify 80 out of 149 peptides that are "dysregulated in abundance" in both conditions; these are presented in Figure 8. Although the authors don't explicitly claim that these peptides are substrates of knocked -down or inhibited PfA-M17, that seems to be the implication. However, inspection of the peptide sequences suggests that the vast majority of dysregulated peptides are unlikely to be PfA-M17 substrates. This is because PfA-M17 has a rather strict specificity, described by the authors as "a strong preference for a P1 leucine and tryptophan residue". Yet only 2 out of 80 peptides possess P1 Leu or Trp, whereas many contain highly polar N-terminal residues (Glu, Gln, Asp, Asn, etc) and are clearly not PfA-M17 substrates. This suggests that the disruption of PfA-M17 is having knock-on effects that cause perturbation of peptide catabolism on a much wider scale, perhaps due to the loss of vacuolar integrity. In addition, it is noted that the experiments are conducted at 10x the EC50, or 3.3 uM. This could result in significant inhibition of PfA-M1, which has an *in vitro* K_i_ of ~4 uM. The obvious incompatibility of the peptides in Figure 6 with the established specificity of PfA-M17, and the possibility of cross-inhibition of PfA-M1, should be discussed in the paper.

Please see the earlier response to this comment. Identification of PfA-M17 as the target of 3 using TPP also used concentrations of up to 12uM and between the two different concentrations, only PfA-M17 was stabilised, indicating that PfA-M1 is unlikely to be inhibited by 3 even at higher concentrations. This has been added to the discussion from line 501.

Lastly, the authors implicate PfA-M17 in amino acid production from hemoglobin by culturing parasites in medium lacking all amino acids but isoleucine, which is not present in human hemoglobin. The idea is that this sensitizes parasites to the diminution of amino acid production from hemoglobin peptides. The authors present data in Figure 9 that suggests that inhibitor 3, but not artemisinin, is more potent when parasites are grown in amino acid-restricted media. These claims are undercut by a technical issue, namely that the inhibitor concentrations used in Figure 9 do not permit the establishment of an upper baseline for the inhibition curves, compromising their reliability. This issue is especially acute for the curves conducted in amino acid restricted media, which do not reach 100% growth.

The inability of the artemisinin data to reach 100% growth when compared to DMSO-treated parasites appeared to be due to the 1 in 2 dilutions from our initial concentration of 1000 nM, which may still be at a high enough concentration to have any effect on growth considering the potency of artemisinin. In order to overcome this, we repeated the artemisinin experiments with a 1 in 3 serial dilution and found parasites to be reaching 100% growth of the DMSO-treated control and observed the same results showing that the use of minimal media has no significant effect on the EC_50_ of artemisinin, as has previously been shown. This is now presented in the lower panel of Figure 10.

Please add a scale bar to Figure 2A.

Scale bar added.

Figure 2C, upper panel, is cropped in an odd manner. It should be cropped such that the band in interest is in the center of the panel.

The initial blots were sectioned so that they could be probed with the different antibodies concurrently, thus unfortunately the upper panel could not be centred. Please see Figure 2- source data 5 and Figure Supplement 2- source data 5 for location of western blot sectioning and probes used.

Supplemental Figure S1 is provided to validate the specificity of the anti-PfAM17 antibody used to generate Figure 2C. However, it looks quite different from the source data file provided (Figure 2-source data 5) which shows at least one major non-specific band (unless this is an alternate form of PfAM17). In the spirit of transparency, it is recommended that this source data file be included in Figure S1.

Figure 2- source data 5 upper panel has been probed with anti HSP-101 (which contains the non-specific, high molecular weight band) that is not relevant for this study. The lower panel is probed with anti-M17 and contains no non-specific bands. Please see Figure Supplement 2- source data 5 for labelled raw data.

Also related to Supplemental Figure S1: this presents data from two separate images (pre-bleed and final bleed) that are merged to give the impression of a single image. This is contrary to best practice; specifically, images that derive from separate blots or exposures should be clearly indicated as such with an intervening space.

This image was one image where the western blot had been cut directly down the centre before being probed with either pre-bleed or final bleed representative of anti-M17 and upon imaging they were placed side by side. To improve clarity we have updated Supplementary Figure 1 which is clearly labelled with the samples, ladder and probes, and source data can be found as Supplementary Figure 1 -source data 1 and Supplementary Figure 1 -source data 1.

"Knockdown of PfA-M17…but not the quantity of hemozoin". This statement seems to be missing a modifier for "quantity of hemozoin".

Amended.

"On analysis of PfA-M17-HAglmS parasites the cycle following knockdown…" This presumably refers to cycle 2 as depicted in Figure 3. This should be explicitly stated to remove any ambiguity.

Amended.

"the main chain amine of Gly489" should presumably read amide.

Corrected.

"Using the structure of 4ZX4.pdf as a template…." What is this structure?

4ZX4.pdb is an existing co-crystal structure of PfA-M1 that we published in 2016 (Drinkwater et al., 2016, Eur J Med Chem). The bound compound is very similar to compound 3 with the exception of the 4-hydroxymethylphenyl group, making it an appropriate template for modelling the position of 3 in the active site of PfA-M1. To clarify this to the reader, we have added the following at line 232 “Using an existing structure as a template (4ZX4.pdb) of PfA-M1 bound to a hydroxamic acid inhibitor that possessed a 3,4,5-trifluorophenyl group rather than the 4-hydroxymethylphenyl found in 3, we were able…”

In Figure 5C, it is not clear what "%growth" refers to. If this is a SYBR green assay, it is sensitive to nucleic acid content, which is not necessarily the same thing as "growth".

Corrected to ‘SYBR Green fluorescence normalised to DMSO treatment’ and scale bar adjusted from 100 to 1.

All of the supplemental figures are at low resolution resulting in a highly pixelated appearance which compromises legibility. Hi-res versions should be provided.

With the original submission low resolution figures were provided due to restraints on Mb that could be uploaded.

Reviewer #3 (Recommendations for the authors):This manuscript offers further characterization of PfA-M17, a *P. falciparum* aminopeptidase that has been studied for some years, and was previously shown to be an essential protein predicted to function in late steps of hemoglobin hydrolysis by erythrocytic parasites. The new report adds valuable demonstration of impacts of PfA-M17 knockdown, description of the synthesis and characterization of a novel PfA-M17 inhibitor with high nanomolar activity against cultured parasites, and a metabolomic analysis of inhibited parasites adding evidence for hemoglobin to be the primary target of the protease. The manuscript is well-written, the described experiments were appropriate, and the conclusions are generally convincing. The manuscript adds importantly to our understanding of the roles of aminopeptidases in parasite biology. However, there are some important concerns, as discussed below.1) Abstract. The statement "these parasites are now resistant to all anti-malaria drugs" is wildly misleading. It is true that resistance is seen to many (but not all; consider lumefantrine and pyronaridine) available antimalarial drugs, but the vast majority of infections are readily treatable with available ACTs. This statement implies otherwise, and it should be changed. There clearly is a need for new antimalarials, but misleading hyperbole to make this claim is unnecessary.

We apologise for this and have amended the statement accordingly so as not to make a misleading claim.

2) Introduction, first paragraph. The WHO dramatically increased estimates for malaria deaths, including revision of past estimates. See the 2021 World Malaria Report. The paragraph should more clearly indicate the role of ACTs in treating falciparum malaria. Artemisinin resistance is NOT "widespread" throughout Asia, but rather to the best of our knowledge confined to a quite small percentage of this large continent, the Greater Mekong sub-region. Limited evidence for spread to India is controversial. Many other parts of Asia have malaria, but without evidence for artemisinin resistance. The statement that artemisinin resistance has recently spread to PNG is both misleading and incomplete. All that is published on PNG is a report on a few isolates with worrisome K13 mutations; it is not clear if this means that resistance is seen in PNG. On the other hand, recent results from Rwanda and Uganda are more convincing, and more worrisome, for spread of artemisinin resistance, and the introduction should better (albeit briefly) summarize the newest data.

We have now amended the introduction to accommodate the concerns of the reviewer and to give a better reflection of what is actually happening in the field.

3) Introduction. "PfA-M17 almost exclusively cleaves leucine and tryptophan *in vitro*" is a bit too simplistic. It would be better for the authors to more accurately, but briefly describe the activity of an aminopeptidase, which is to cleave AAs from the amino-terminal end of peptides or proteins. Indeed, this is provided, but in the middle of the Results section rather than in the Introduction, where it will be more useful for readers.

This was provided in the introduction, but we have reworded it to make it clearer what the function of aminopeptidases are (see line 87-89 and lines 117-121).

4) Results, first paragraph. The relevance of this discrepancy is unclear, but it is misleading to state that the HA tagged PfA-M17 construct was seen on a Western blot at "slightly lower than the predicted 72 kDa size". In fact the size appears to be much lower than 72 kDa, and the authors should discuss potential explanations for this surprising result.

We have now included some comments at line 153-158 in regard to what could cause this discrepancy in size, which could include proteolytic cleavage of proteins upon lysis of parasites given PfA-M17 harbours a low complexity region at its N-terminal end. We have also added that Supplementary Figure 1 shows a distinct molecular mass increase between Pf3D7 wild type PfA-M17 and triple HA-Strep tagged protein, indicating that while the PfA-M17 size is smaller than expected, it is indeed recognising PfA-M17 in PfA-M17-HAglmS parasites given that we have shown correct integration of the tag.

5) Results, P. 8. The following subtitle has uncertain meaning, and should be rewritten: "Knockdown of PfA-M17 results in multiple digestive vacuoles forming during trophozoite stage but not the quantity of hemozoin".

Corrected to ‘Loss of PfA-M17 results in the formation of multiple digestive vacuoles’.

6) Figure 4A. The demonstration of multiple DVs is interesting and potentially very important in terms of characterizing mechanisms, but the micrograph showing this phenotype is sub-optimal. A larger, sharper image, and ideally multiple images showing this phenotype will be helpful. The figure suggests that only a small minority of parasites had >1 DV; quantification of this result (what percentage of control and treated parasites had >1 DV?) would be helpful.

On reflection we agree that it was incorrect to label hemozoin fragmentation as multiple digestive vacuoles and have amended this throughout. We have additionally performed transmission electron microscopy on 3 treated parasites where it can be seen that parasites have multiple membrane bound digestive vacuoles each containing hemozoin. This is in comparison to vehicle control treated parasites which contain only a single digestive vacuole. The percentage of parasites that contain multiple hemozoin fragments has been added to the Results section, where approx. 30% of knockdown parasites contained fragmented hemozoin and approx. 40% of 3 treated parasites had fragmented hemozoin. These results are discussed under the heading ‘Loss of PfA-M17 results in the formation of multiple digestive vacuoles’.

7) Results, P. 12-13. The metabolomics experiments are elegant, but the argument derived from these studies is somewhat unsatisfying. Inhibition of an aminopeptidase is expected to result in accumulation of peptides. But, after knockdown or treatment with an inhibitor of the aminopeptidase, is this accumulation specifically of Hb-derived peptides? The results are suggestive, but the peptides shown to accumulate were all short (2-5 AAs), so identification as Hb-derived is uncertain; many might similarly be seen as hydrolysis products of any large protein. Can the authors better justify their argument, perhaps with a simple mathematical explanation for why assignment of this number of short peptides as Hb products is strongly suggestive of Hb peptides as the natural targets of PfA-M17? In other words, how sure can we be that assignment of a number of short peptides to hemoglobin is not simply due to chance?

To more directly link the dysregulated peptides to haemoglobin, we have acquired additional MS/MS data from which we derive accurate sequences (as opposed to residue composition inferred from total peptide mass) for 43 peptides. Further, we have counted the number of times dysregulated peptides can be mapped to any of the ~4700 Plasmodium and human proteins detectable in infected erythrocytes. These counts, normalised by protein length, provide a measure of how similar a given protein is to the set of dysregulated peptides. By this measure, the Hb chains α, α2, β, and δ are four of the five most-similar proteins. Please refer to the results from line 384 to 397.

8) P. 14. Artemisinin should not be capitalized.

Corrected.

9) Discussion. The discussion offers a nice discussion of the relevance of different substrate specificities of different plasmodial aminopeptidases. The report offers a detailed list of peptides that accumulate with knockdown or inhibition of PfA-M17. Are these peptides consistent with loss of the specific proteolytic specificity of PfA-M17 (with preference for P1 leucine and tryptophan residues)?

Indeed, we agree that the broad impact of PfA-M17 depletion/inhibition is somewhat surprising, as it is not merely limited to Leu/Trp peptides. Whilst it is possible that PfA-M17 has a broader substrate specificity within the intracellular context than what has been demonstrated in simplified enzyme assays, the extensive dysregulation (~80 peptides) observed here suggests a secondary impact on proteolytic processes. The effect may be due to direct inhibition of other proteases/peptidases by the accumulation of PfA-M17 substrates, dysfunctional digestive vacuole activity associated with the observed morphology, and/or secondary dysregulation of proteolytic enzyme levels or functions. Some comments about these potential effects have been added to the discussion at lines 484-500.

10) Discussion. Some additional discussion should address the presentation of PfA-M17 as a potential drug candidate. What is the likelihood that improved potency can be engineered, as the current lead offers relatively modest (high nanomolar) *in vitro* potency? What is the evidence that compound 3 does not inhibit host proteases? Are there concerns about the quite slow action of the inhibitor against cultured parasites?

This compound serves as an excellent tool for elucidating the function of PfA-M17 and our results show that PfA-M17 is a potential drug target. The TPP indicated that only parasite PfA-M17 is the target. We have added in the discussion that while human leucyl aminopeptidases are currently poorly characterised, it will be important for future PfA-M17 compounds to be tested for off-target effects. In keeping with our knockdown results, 3 treatment results in a delay in parasite growth, however it is most active against trophozoite stage parasites so any treatment at ring stages is less effective.

BIRRELL, G. W., CHALLIS, M. P., DE PAOLI, A., ANDERSON, D., DEVINE, S. M., HEFFERNAN, G. D., JACOBUS, D. P., EDSTEIN, M. D., SIDDIQUI, G. & CREEK, D. J. 2020. Multi-omic characterization of the mode of action of a potent new antimalarial compound, JPC-3210, against *Plasmodium falciparum*. Mol Cell Proteomics, 19, 308-325.

CREEK, D. J., CHUA, H. H., COBBOLD, S. A., NIJAGAL, B., MACRAE, J. I., DICKERMAN, B. K., GILSON, P. R., RALPH, S. A. & MCCONVILLE, M. J. 2016. Metabolomics-based screening of the Malaria Box reveals both novel and established mechanisms of action. Antimicrob Agents Chemother, 60, 6650-6663.

GIANNANGELO, C., SIDDIQUI, G., DE PAOLI, A., ANDERSON, B. M., EDGINGTON-MITCHELL, L. E., CHARMAN, S. A. & CREEK, D. J. 2020. System-wide biochemical analysis reveals ozonide antimalarials initially act by disrupting *Plasmodium falciparum* haemoglobin digestion. PLoS Pathog, 16, e1008485.

HARBUT, M. B., VELMOUROUGANE, G., DALAL, S., REISS, G., WHISSTOCK, J. C., ONDER, O., BRISSON, D., MCGOWAN, S., KLEMBA, M. & GREENBAUM, D. C. 2011. Bestatin-based chemical biology strategy reveals distinct roles for malaria M1- and M17-family aminopeptidases. PNAS, 108, E526-E534.

OKOMBO, J., KANAI, M., DENI, I. & FIDOCK, D. A. 2021. Genomic and genetic approaches to studying antimalarial drug resistance and Plasmodium biology. Trends Parasitol, 37, 476-492.

[Editors’ note: what follows is the authors’ response to the second round of review.]

The manuscript has been improved but there are some remaining issues that need to be addressed, as outlined below:Both reviewers concur that the TPP data need to be clarified and these revisions should be made. No new experimentation should be required for that.Reviewer #1 (Recommendations for the authors):The authors have substantially revised the manuscript in response to the prior review and as a result the claims in the manuscript are more robust. The authors have provided an impressive quantity of new data; notable examples include the analysis of fragmented food vacuoles by TEM, MS/MS analysis of key peptide metabolites in the metabolomics experiment and analysis of their probable origin from globin, and attempts to establish the specificity of compound 3 by a variety of approaches. Overall, this is a very solid manuscript that convincingly establishes the essentiality of PfA-M17 and its role in peptide catabolism.The only remaining concern relates to the newly-added thermal proteomics profiling (TPP) study. In this approach, addition of a small molecule ligand to a proteome is expected to specifically stabilize the target against thermal denaturation. The authors conducted TPP at two concentrations of compound 3 and present these data in Figure 6. They claim that "only a single protein was significantly stabilized across the two drug concentrations…PfA-M17". However, looking at Figure 6A, it is clear that there are at least a dozen proteins that are more highly stabilized in both drug concentrations. Is the implication that none of these proteins was "significantly stabilized"? If so, it doesn't make sense on an intuitive level that all of the proteins in that exhibit a higher fold-change are not significant, whereas PfA-M17, which at the edge of the undifferentiated mass of "unchanged" proteins, is significant. To allow readers more insight into this study, the authors should provide (as supplemental data) the identities of the proteins in the upper right quadrant of the 3 µM fold change vs. 12 µM fold change plot (about 15-20 proteins), along with relevant mass spec data (number of unique peptides, #s of technical replicates in which they are observed) and the fold-change values and associated statistics. It would also be helpful to include the data for PfA-M1 as this is the most likely off-target. This would help to put the claims of on-target binding of compound 3 on a solid foundation.

We thank the reviewer for the comments and the results (line 308-319), methods (line 887-892), Figure 6 and Figure 6 legend (line 1205-1208) have now been amended to better clarify our TPP data. We have also included this data in the source data file (consisting of intensity and peptide count of proteins identified across replicates, with fold change compared to DMSO control and associated P-value). We have highlighted both *Pf*A-M17 and *Pf*A-M1 in yellow in the excel sheet with PfA-M1 bolded to show that *Pf*A-M1 thermal profile was not affected in both drug concentrations following heating at 60°C.

The authors should also provide more details on the "LC-MS/MS and data analysis" methods related to TPP. The statement "Following protein identification and quantification…" needs to be elaborated on, as this is the most important set of experimental details related to understanding Figure 6. Things that would be useful to know: was there a minimum unique peptide cutoff? How many peptides from a given protein were used in quantitation? Did these have to appear in all technical replicates? Data in Figure 6B are reported as "Protein relative abundance"; relative to what?

For the methods section, please see above, this has now been amended to include the relevant information and the source data is provided, while Figure 6B has been modified. Yes there was a minimum peptide cutoff and they had to appear in all technical replicates.

Reviewer #2 (Recommendations for the authors):In this revised version of their manuscript, the authors have dealt satisfactorily with the majority of the issues raised in response to the original submission. In particular, the use of TEM to better visualise the 'multiple DV' phenotype resulting from PfA-M17 knockdown adds significantly to the study, and the thermal proteomic profiling (TPP) analysis helps substantiate Pf-M17 as an in vivo target of compound 3. These additions to the work are appreciated and definitely improve the manuscript. However, there is still a lack of clarity in some sections of the manuscript, generally due to inaccurate language.Requested modifications:Lines 309-311 and Figure 6 – This part of the manuscript describes the TPP analysis. It is stated in the text that '…only a single protein was significantly stabilised across the two drug concentrations…'. Furthermore, a similar statement is made in the Figure 6 legend, where it is stated that: '…a single protein (Pf3D7_1446200) was significantly altered at both concentrations…'. However, the volcano plots shown in Figure 6 show that in fact several proteins (~12 and ~14 respectively) fall into the right-hand top quadrant of the plots at drug concentrations of 3 μM and 12 μM, suggesting that these are stabilised by drug treatment too. Do the authors mean that PfA-M17 was the only protein that was in common between these two protein subsets showing increased stability at the two drug concentrations used? If so, they should specifically and more accurately state this.

Yes, this is what we mean. The text has now been amended to better clarify our TPP data and we have also included this data in the source data file to aid with the understanding of our data set. (See response to Reviewer 1)

Lines 1222 and 1228 (Figure 7 legend) – replace 'medium' with 'median'? Alternatively, explain these terms better.

Corrected.

Supplementary Figure 2A and 2B: Please indicate the concentration of glucosamine used in both the figure legend and the figure panels.

The figure legend for amended Figure 3—figure supplement 2 has been edited to indicate parasites were treated with 2.5mM where appropriate. Panel C contains the appropriate glucosamine concentration added.

Supplementary Figure 3: these data were included in response to the suggestion to include a full microscopic field of the control and glucosamine-treated Pf-M17-HAglmS parasites, in order to demonstrate depletion of the HA-tagged protein at the single parasite level following treatment with glucosamine. Unfortunately, the provided figure is still sub-optimal; it should include panels in which the same parasites are probed with a control antibody (i.e. specific for an irrelevant parasite protein), as well as DIC or bright-field images to show the morphology of the parasites. Please provide a suitable replacement image.

Figure 3—figure supplement 1 has now been replaced with a figure where parasites have been probed with anti-HA (to detect *Pf*A-M17-HA) as well as anti-EXP2, a parasitophorous vacuole membrane (PVM) protein, the latter of which the labelling is unchanged in parasites treated with 2.5mM glucosamine. Merged images including a bright field are now included. We extend our apologies for the mark on the bright field due to a faulty camera.